# Planning the electric vehicle transition by integrating spatial information and social networks

Jiaman Wu [1,2], Ariel Salgado[2,3] & Marta C. González [2,4,5] ✉

The transition from gasoline-powered vehicles to plug-in electric vehicles (PEVs) offers a promising pathway for reducing greenhouse gas emissions. Spatial forecasts of PEV adoption are essential to support power grid adaptation, yet forecasting is hindered by limited data at this early stage of adoption. While different model calibrations can replicate current trends, they often yield divergent forecasts. Using empirical data from states with the highest levels of adoption in the United States, this study shows that accounting for spatial and social networks among potential PEV adopters produces forecasts that are only one-third of benchmark predictions for 2050. Results further demonstrate that incorporating spatial social networks improves the ability to capture the spatial autocorrelation observed in the empirical diffusion process. This study also evaluates the potential impact of various PEV marketing campaigns under prevailing uncertainties, highlighting the importance of tailoring strategies to network dynamics for effective PEV promotion.

Modern techno-social systems consist of large-scale physical infrastructures embedded in a complex communication web whose dynamics are driven by human behavior[1]. This interplay is illustrated by the growing integration of plug-in electric vehicles (PEVs)−including both battery electric vehicles (BEVs) and plug-in hybrid vehicles (PHEVs)−into power systems. PEVs offer substantial environmental advantages by reducing traffic-related air pollutants and cutting down greenhouse gas emissions[2,3]. In addition, they provide economic benefits such as improved fuel efficiency and enhanced energy security[4]. Recognizing these advantages, the United States has enacted policies to encourage the uptake of PEVs. In 2022, the California Air Resources Board (CARB) approved the Advanced Clean Cars II regulation to ensure that by 2035, all new cars and light trucks sold in California shall be zero-emission vehicles (ZEVs)[5]. That same year, Washington state announced it would follow California and prohibit the sale of new gas-powered vehicles by 2035[6]. Meanwhile, worldwide efforts to accelerate the adoption of PEVs are exemplified by the Chinese mandate for new energy vehicles[7], the United Kingdom's plan to transition to ZEVs[8], and

the European Union's $CO_2$ emissions regulations[9]. These initiatives are expected to substantially boost adoption in the near future. Aligned with this trend, modeling the geographical distribution of future PEV adopters becomes a pivotal task. This is because human decisions on when and where to adopt PEVs are linked directly to the regional electricity demands[10], the consequent pressure on the power grid[11], and the power system infrastructure planning[12]. Computational modeling approaches can provide predictability in complex techno-social systems if fed with the right data. Yet, modeling the spatial distribution of future PEV adopters is challenging. Despite increasing share in global vehicle sales[13], the adoption rate of PEVs is low[14], and with high regional disparities globally as well as within a country[15]. This early stage of adoption results in limited data to forecast the geographical distribution of PEV adopters.

The diffusion of PEVs exhibits great differences across different demographic groups at the early stage. Evidence from the global PEV market indicates that early PEV adopters tend to have higher incomes[16], advanced education[17], and reside in non-apartment houses

[1]Department of Data Science, City University of Hong Kong, Hong Kong SAR, China. [2]Department of Civil and Environmental Engineering, UC Berkeley, Berkeley, CA, USA. [3]Instituto de Cálculo, UBA-CONICET, Ciudad de Buenos Aires, Argentina. [4]Department of City and Regional Planning, UC Berkeley, Berkeley, CA, USA. [5]Lawrence Berkeley National Laboratory, Berkeley, CA, USA. ✉e-mail: martag@berkeley.edu

with home charging access[18]. In this regard, forecasting PEV adoption involves addressing the complex challenge of modeling the impact of various factors on adoption across diverse demographic segments. These factors can be broadly categorized as innovation-related and imitation-related. Innovation-related factors determine the extent to which consumers are inclined to adopt PEV without the impact of social influences, whereas imitation-related factors pertain to word-of-mouth effects from individuals who have already adopted or have experience with the PEV. Specifically, innovation is relevant to vehicle-specific attributes such as purchase cost and maximum driving range[19], mass media or advertising regarding PEV[20], the built environment such as the availability of public charging infrastructure[18] and grid infrastructure[21], availability of monetary incentives such as rebate upon purchase and tax reduction after purchases[22], and non-monetary incentives such as high-occupancy vehicle (HOV) lane access for adopters[23]. Imitation, on the other hand, is shaped by social influences from friends[24], online interactions[25] as well as neighborhood and workplace interactions[26,27].

Previous studies establish a connection between the aforementioned factors and the PEV diffusion projection utilizing aggregated and disaggregated data. The aggregated data is represented by the share of PEV vehicle sales at the country level[28], whereas the disaggregated data includes individual purchase records[29,30] and surveys on travel behavior[31] or home charging access[32]. The spectrum of methodologies ranges from discrete choice models[33] to regression models[34]. For instance, Forsythe et al.[35] conduct a discrete choice experiment among new vehicle consumers in the United States. This study indicates that projected advances in BEV range and price may allow them to equal or exceed the consumer valuation of gasoline vehicles by 2030. Wu et al.[36] employ Bayesian models to predict individual adoption probabilities and map the spatial distribution of charging demand of prospective adopters under varying adoption scenarios in the San Francisco Bay Area. They demonstrate that personalized shifting recommendations regarding charging schedules can reduce charging demand during power grid peak hours by 61% when achieving 90% PEV stock. Bayesian models can project the geographical distribution of future adopters given any adoption rate. However, it cannot output the time to reach such adoption. To provide temporal projections, the Bass model (BM) is widely used. The BM analyzes the market as a whole using two parameters (related to imitation and innovation factors, respectively). In 2009, Becker et al.[37] incorporated assumptions related to oil price fluctuations and incentives with BM and projected that PEVs could represent 65% of vehicle sales by 2030. Jenn et al.[38] utilize sales data from 39 countries and explore PEV market expansion through BM. They find that it may be challenging to achieve targets such as 100 million electric vehicles worldwide by 2030. Lee et al.[39] use BM to predict the adoption dynamics of heterogeneous groups of PEV buyers. Their findings indicate that although high-income families currently constitute 49% of the PEV market, they form only 3.6% of households in California, suggesting this group may not remain the dominant PEV adopters. To sustain market growth, policymakers should account for the infrastructure and incentive requirements of consumers from mid/high- and middle-income groups. Despite the ability to provide temporal projections, BM does not provide the spatial distribution of future adopters. One limitation of the BM lies in the fact that the formulation of the BM implies homogeneity and global interconnectedness, i.e., each agent's probability of adoption is influenced uniformly by the adoption state of all other agents[40]. This leads to the omission of spatial information and characterization of heterogeneity in social influence. Another limitation of modeling adoption with the BM is that the model represents each demographic group in isolation and thus disregards any interaction the groups may have. To incorporate the role of geography and demographic characteristics in diffusion, we need models that consider both the spatial distribution of individuals and heterogeneity in the social network connections among different socio-demographic groups.

In this work, we investigate how spatial social networks influence PEV adoption forecasts by comparing benchmark adoption models at different spatial scales. We first aggregate individual purchase records at state, county, and census tract levels, and then we fit the empirical adoption trends at different levels of spatial aggregation in Washington and California. To that end, we consider two versions of the BM: the original BM and the Social Network BM (SocNet BM). Results reveal that both models can be calibrated to fit data up to 2022, but including or excluding social network effects leads to a threefold difference in adoption rate projections by 2050. Additionally, we demonstrate that the SocNet BM allows more flexibility to capture the spatial auto-correlation of the empirical diffusion process than the State/County BM. This spatial information is critical for planning, charging, and grid infrastructure. At last, we point out that promotion campaigns should be designed based on the understanding of spatial social network structure in the area of interest, as the effectiveness of campaigns is tied to the local socio-demographical structure. Altogether, the study offers insights into the forecast based on early adopters that is valuable to planning power grid demand and the allocation of charging infrastructure.

## Results
### Adoption overview

California and Washington have the greatest market share of PEVs in the United States by the end of 2022[41]. In this section, we use PEV purchase records in these two leading states for the adoption overview. The Clean Vehicle Rebate Project (CVRP) provides individual electric vehicle purchase records with adopters' home census tract and the purchasing date in California between 2010 and 2022[29]. The Electric Vehicle Title and Registration Activity (EVTRA) includes records of PEV ownership with adopters' home census tract and the purchasing date in Washington between 2010 and 2022[30]. We get data on the number of passenger vehicles and household median incomes at the census tract level between 2010 and 2022 from the Census Bureau[42].

We empirically investigated the PEV adoption process from 2010 to 2022. On the state level, Fig. 1A shows that the total adoption rate in Washington is around 1.9% in 2022, and the incremental adoption rate presents an increasing trend from 2010 to 2022. Figure 1D shows that California has a total adoption rate of around 2.0% in 2022. The incremental adoption rate reached local maxima in 2017. This may have happened due to rebate program policy changes at the end of 2016[43]. Following the Diffusion of Innovation Theory[44], Rogers divides adopters as (1) innovators: first 2.5%, (2) early adopters: next 13.5%, (3) early majority: following 34%, (4) late majority: next 34%, and (5) laggards: last 16%. This locates current state-level adoption trends in the innovators' phase.

On the county level, Fig. 1B shows that most PEV adopters from Washington live in King County and Pierce County. Figure 1E shows that most PEV adopters from California live in the Bay Area and Los Angeles County. These two maps indicate that PEV adoption started in metropolitan areas with a higher population and has not yet reached more rural areas. On the census tract level, we examine the correlation between socio-demographic characteristics and the total adoption rate. Figure 1C, F show, respectively, the adoption rate and the corresponding median household income of each census tract in Washington and California in 2022. Fitting the relationship between the adoption rate and the corresponding median household income results in similar slope coefficients of 0.04 ($R^2 = 0.52$) and 0.03 ($R^2 = 0.50$) for Washington and California. According to the correlation between adoption rate and income, we segment the potential market into low-, middle-, and high-income groups for adoption modeling purposes (see the section "Market segmentation").

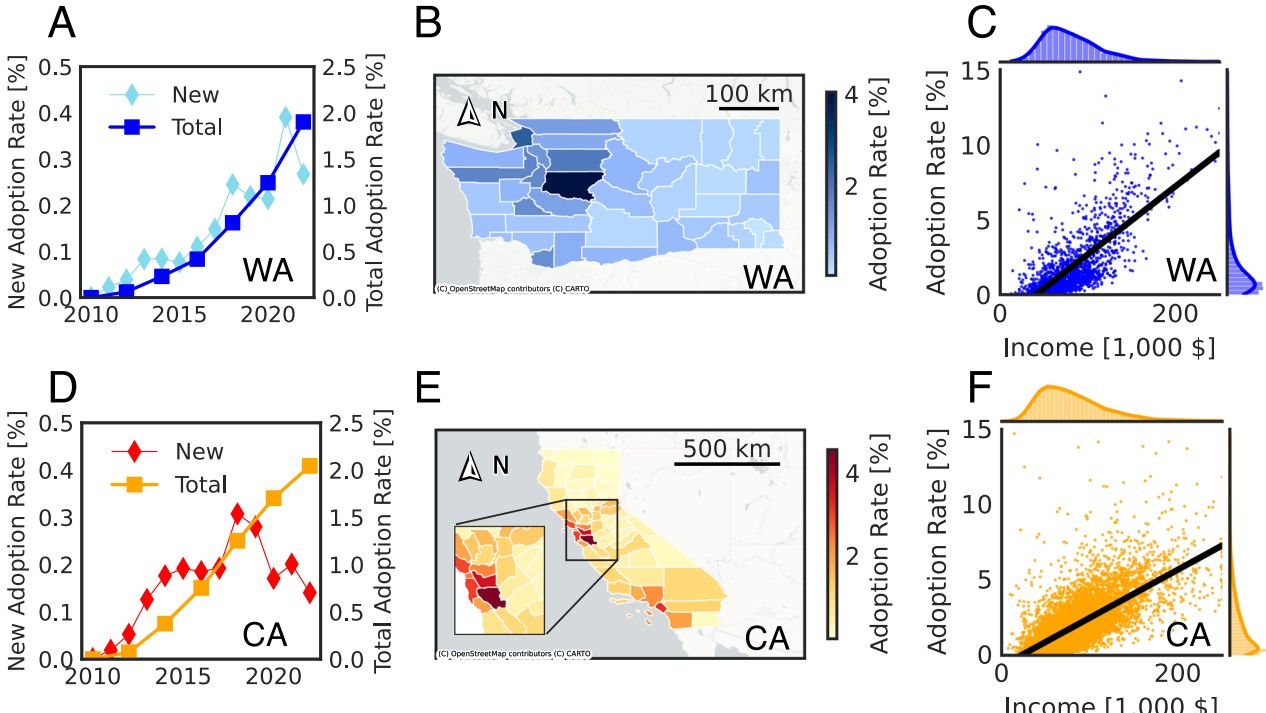

Fig. 1 | **Overview of current PEV adoption. A**, **D** State-level yearly total adoption rate (circle marker) and new adoption rate (square marker) in Washington (blue) and California (orange) from 2010 to 2022. **B**, **E** Maps of county-level cumulative adoption rate in Washington (blue) and California (orange) in 2022. A zoom-in adoption rate map in the Bay Area is attached with **E**. **C** and **F** Correlation between the adoption rate and the corresponding median household income of all census tracts in Washington (blue) and California (orange) in 2022. The black regression line, given the adoption rate and the corresponding median household income, is shown with points.

## Calibration of PEVs diffusion models

We use the BMs at the state level and the county level as benchmarks, referred here as State/County BM (see the section "Benchmark BM"). In BMs, the probability of every agent adopting, given that it has not adopted so far, depends linearly on an independent innovation-related influence $p$ and an imitation-related influence $q$ that depends on the fraction of prior adopters[45]. This formulation implies homogeneity and global interconnectedness and fails to capture the heterogeneity in social connections among different socio-demographic groups from different locations.

Here, we further investigate the adoption spreading on a social network embedded with geographical and socio-demographic information via the SocNet BM (see the section "SocNet BM"). We start by creating social networks preserving preferential attachment connectivity and distance effects features following[46]. In a network, the concept of preferential attachment suggests that the likelihood of forming a connection with a node increases with its number of connections. Meanwhile, geographical effects indicate that nodes are more inclined to form connections with their geographically close neighbors. Therefore, in a network incorporating both preferential attachment and geographical effects, any long-distance connections that do form are typically directed toward highly connected nodes. This suggests a dual influence on network growth: while local connections favor spatial proximity, distant connections tend to target nodes with high connectivity, creating a network characterized by both local clustering and scale-free properties[46]. With social networks, we fit SocNet BM parameters, namely $p$ and $q$ for low-, middle-, and high-income groups with empirical county-level aggregated PEV sales data (see the section "SocNet BM". Considering that varying $p$ and $q$ can fit the data at this early stage (we defer detailed explanation in the section "Effects of network structure on spatial diffusion"), we repeat the fitting process and report the median and uncertainty interval of

results in this study. Afterward, we compare how well the State/County BM and SocNet BM fit the data by investigating fitting errors at the state, county, and census tract levels.

Figure 2A and B show the state-level aggregated adoption rate in Washington and California between 2010 and 2022. All models fit similarly well with the empirical data, i.e., State/County BM has a mean absolute error of 0.03% and SocNet BM of 0.08% in Washington. In California, the errors are 0.08% and 0.11%, respectively. Different from County BM and SocNet BM, which directly output county-level forecasts, State BM fits and projects state-level adoption rates. To get the county-level aggregated adoption rate from State BM, we simulate the PEV adoption growth curve on the state level based on fitted parameters and assign the PEV adopters to census tracts proportionally to the population. Figure 2D and E show the county-level aggregated cumulative adoption rate in Washington and California in 2022. We present the 15 most populated counties in each state and show the aggregated cumulative adoption rate for other counties in Supplementary Information (see Supplementary Information Note S1). We find that County BM and SocNet BM fit the data similarly well. However, State BM underestimates adoption rates in more populated counties and overestimates adoption rates in less populated counties. For example, it underestimates the adoption rate in King County while overestimating the adoption rate in other counties in Washington. In California, the State BM underestimates the adoption rate in Los Angeles, Orange, Santa Clara, and Alameda County while overestimating the adoption rate in other counties.

On the tract level, Fig. 2C shows the distribution of adoption rate error for three models. The adoption rate error values are the absolute difference between the simulated adoption rate and the ground truth adoption rate in 2022. The errors for most tracts in Washington and California are within 2.5%. The tract-level error of SocNet BM has a higher variance than State/County BM (see Supplementary

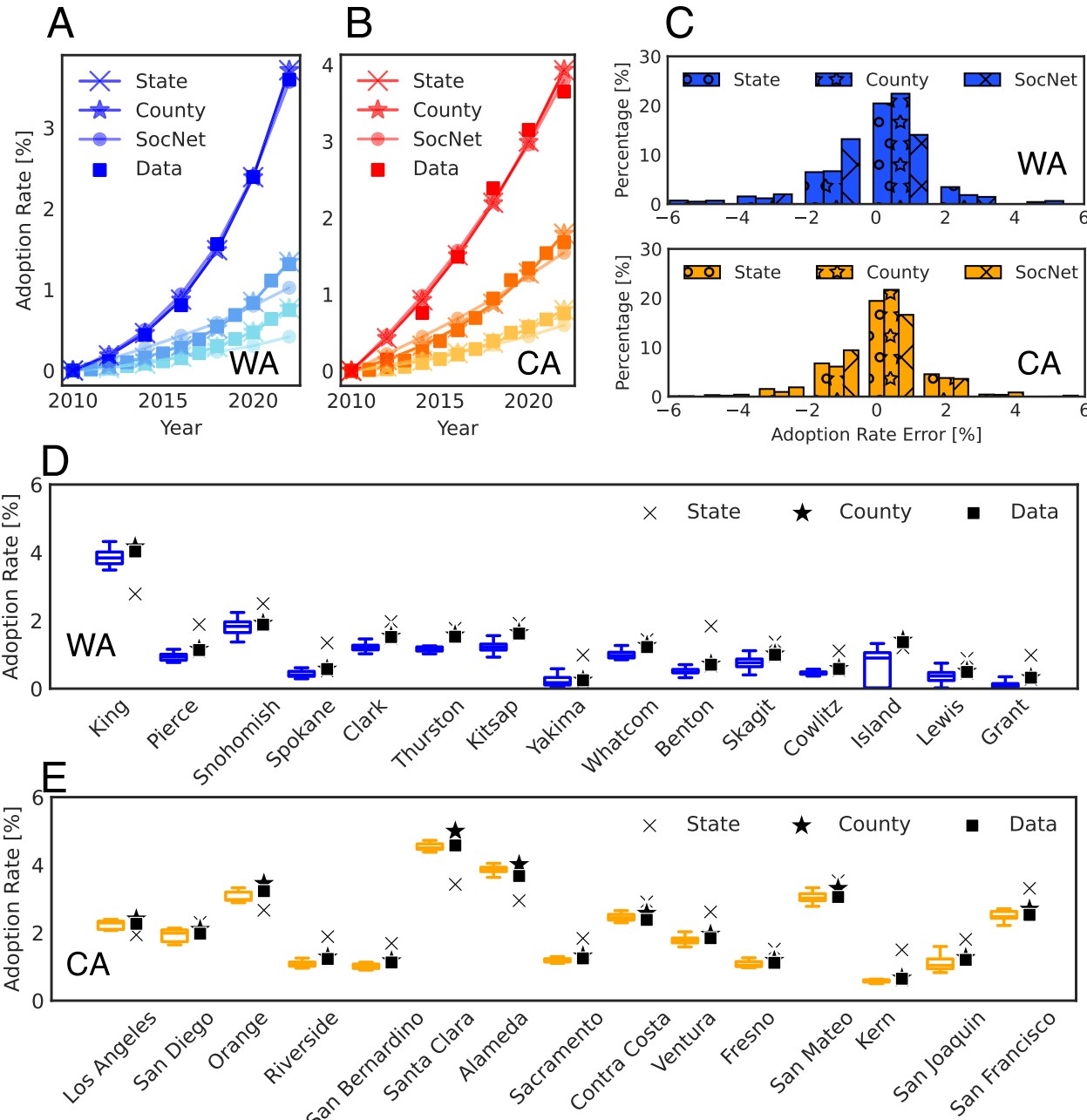

**Fig. 2 | Comparison of State/County BM and SocNet BM's performance.** State/County BM is based on a uniquely estimated parameter set, while SocNet BM results are obtained by averaging 10 simulations for each of three independently estimated parameter sets. **A** and **B** State-level fitted adoption curves of State BM (cross), County BM (star), SocNet BM (circle), and data (square) for low (light blue/yellow), middle (blue/orange), and high (dark blue/red) income groups in Washington and California from 2010 to 2022. Solid lines with markers show the median (50th percentile) adoption rate of income groups in states. **C** Distribution of tract-level adoption rate error of State BM (cross), County BM (star), and SocNet BM (circle) in Washington (blue) and California (orange) in 2022. **D** and **E** County-level adoption rates of State/County BM and SocNet BM in the most populated 15 counties in Washington (blue) and California (orange) in 2022. The box represents the interquartile range (25th–75th percentiles) of county-level adoption rates of SocNet BM, the center line marks the median, and the whiskers indicate the minimum and maximum values. Cross markers show the adoption rate of State BM, star markers show the adoption rate of County BM, and square markers show the empirical data.

Information Note S2) since (1) it characterizes the adoption diffusion on an individual basis, while State/County BM uses only three $p$ and $q$ sets (fitted from three socio-demographic groups) to characterize the adoption diffusion. (2) It characterizes the uncertainty in the adoption process by simulating it in an agent-based and probabilistic manner.

### Parameters of PEVs diffusion models

The innovation parameter $p$ reflects the extent to which consumers are willing to adopt new products without being affected by others. It is widely recognized that vehicle-specific attributes[19], mass media[20], the built environment[18,23], and availability of incentives[22] are relevant to the willingness of consumers to accept PEVs. However, to what extent these factors influence whether people adopt PEV remains an open question. The imitation parameter $q$ describes the "word-of-mouth" effect[24–27]. It captures how adopters are affected by the number of adopters in their social network. In general, a higher $p$ indicates that the new product is likely to attract more early adopters, whereas a higher imitation coefficient is associated with faster acceleration of diffusion in later periods.

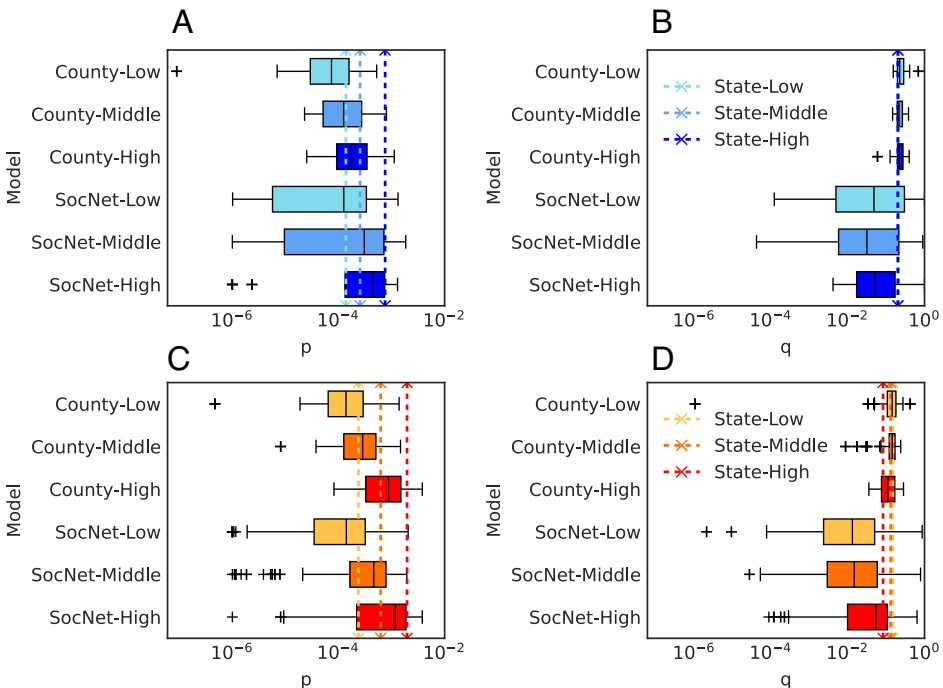

**Fig. 3 | Calibrated parameters for socio-demographic groups using State/County BM and SocNet BM in Washington (blue) and California (orange).** State/County BM is based on a uniquely estimated parameter set, while SocNet BM results are obtained by three independently estimated parameter sets. The box represents the interquartile range (25th–75th percentiles) of the county-level $p$ and $q$ parameters of County BM and SocNet BM, the center line indicates the median, the whiskers extend to the minimum and maximum values within the distribution, and '+' marks denote outliers. The vertical dashed line denotes the $p$ and $q$ of State BM. The color denotes different socio-demographic groups, i.e., light color for the low-income group, default color for the middle-income group, and dark color for the high-income group. **A** and **B** $p$ and $q$ parameters of State/County BM and SocNet BM for low, middle, and high-income groups in Washington. Each box represents the distribution of results across 58 counties from three estimated parameter sets. **C** and **D** $p$ and $q$ parameters of State/County BM and SocNet BM for California's low, middle, and high-income groups. Each box represents the distribution of results across 58 counties from three estimated parameter sets.

We show the distribution of obtained parameters for State/County BM and SocNet BM for the three socio-demographic groups in Washington and California in Fig. 3 and attach median values of these parameters in Supplementary Information (see Supplementary Information Note S3). Figure 3A shows that in Washington, the high-income group has the highest $p$ and the low-income group has the lowest $p$ for three models. Figure 3B shows that the $q$ for State/County BM is similar across the three groups in Washington. Similar to Washington, the high-income group in California has the highest $p$ and the low-income group has the lowest $p$ as shown in Fig. 3C. Compared with Washington, California has a higher $p$ for the three income groups at the state level. Figure 3D shows that in California, the $q$ of the State/County BM and SocNet BM are similar across three groups. The difference between State/County BM and SocNet BM is that the fitted parameters of SocNet BM have more variance. This variance is brought by fitting with limited data, and also the probabilistic nature of the agent-based simulation process.

### Divergence of PEVs adoption forecasts

We forecast PEV adoption rates from 2023 to 2050 with the calibrated models. In this section, we present the most optimistic adoption forecast by assuming that PEV users do not revert to internal combustion engine vehicles (ICEVs). The forecast considering the potential switching back to ICEVs by PEV users is deferred to Supplementary Information (see Supplementary Note S4). Figure 4 shows the comparison of State/County BM and SocNet BM's prediction. Interestingly, we find that the social network model is three times more conservative in future forecasts, even though the three models all fit the empirical data well (as shown in Fig. 2).

Figure 4A, C show that the adoption growth curves of State BM and County BM are similar, as they all assume global interconnectedness in social network[40]. In contrast, SocNet BM has a much lower adoption rate prediction than State/County BM. This is because the SocNet BM uses a social network model, while individuals are influenced by the whole population in the State/County BM. Figure 4B, D demonstrates the predicted county-level adoption rate in 2050 for the three considered models. We present the most populated 15 counties in Washington and California and show other counties in Supplementary Information (see Supplementary Information Note S5). Although State BM and County BM are similar when aggregated at the state level, they are very different at the county level. For instance, the adoption rate of King County using State BM is lower than using County BM, while Pierce County using State BM is higher than using County BM. Similarly, the adoption rate of Orange County using State BM is lower than using County BM, while Santa Clara County using State BM is higher than using County BM. This shows that when looking at a smaller geographical scale, the predictions are very different, even if they show similar results on an aggregated level. In contrast to State/County BM, the prediction of SocNet BM has a much higher variance. This is due to two reasons. First, different combinations of $p$ and $q$ parameters give the same good fit with empirical data, but could be very different as time increases. Second, even with the same $p$ and $q$ parameters, the probabilistic agent-based model itself brings variance to the prediction.

### Effects of network structure on spatial diffusion

Knowing the spatial distributions of PEV adopters is important for charging station siting and grid capacity assessments. Since spatial information is embedded in the construction of the social networks, we test various network parameters $\gamma = 1, 3, 10$ for Los Angeles, where $\gamma$ is a parameter of the distance selection function (see the section "SocNet BM"). Higher $\gamma$ favors links between nodes with a smaller distance. To further investigate the spatial characteristics of the

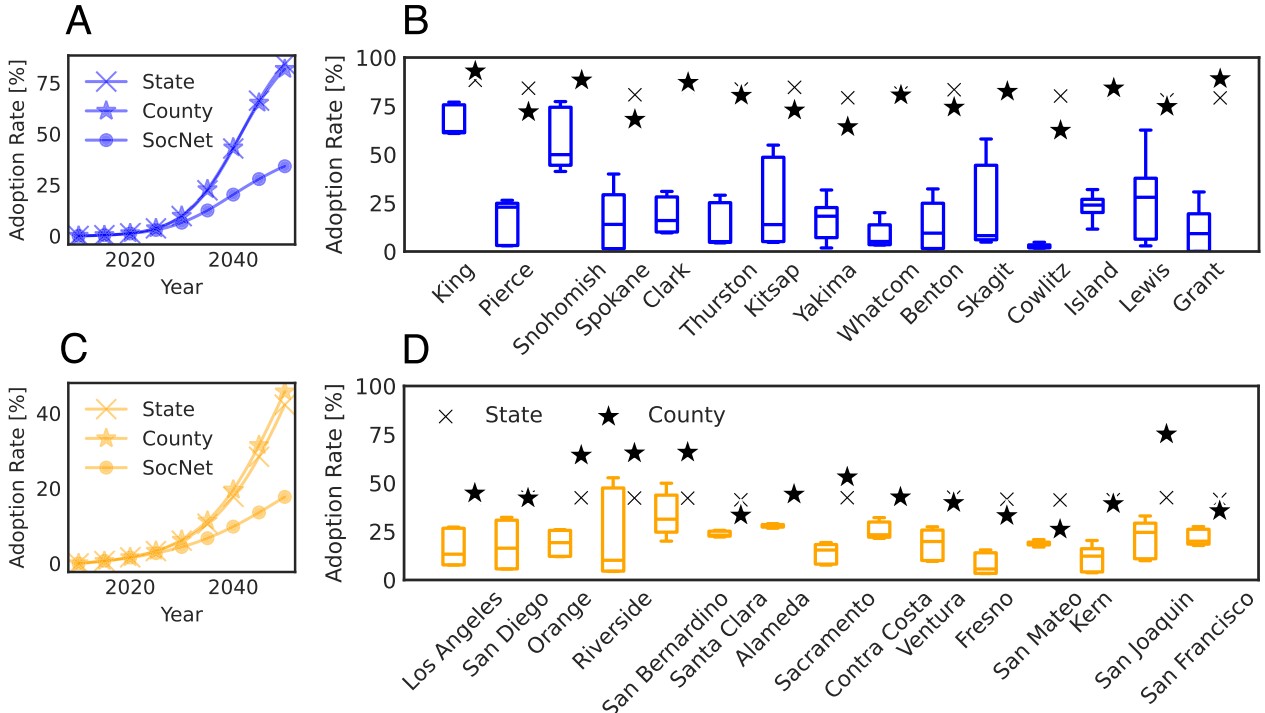

**Fig. 4 | Comparison of State/County BM and SocNet BM's adoption rate prediction.** State/County BM is based on a uniquely estimated parameter set, while SocNet BM results are obtained by averaging 10 simulations for each of three independently estimated parameter sets. **A, C** State-level adoption rate growth curve from 2010 to 2050 for State/County BM and SocNet BM in Washington (blue) and California (orange). Lines with cross markers show the state-level adoption rate of State BM. Lines with star markers show the state-level adoption rate of County BM. Lines with circle markers show the median (50th percentile) state-level adoption rate of SocNet BM. **B, D** The county-level adoption rate from State/County BM and SocNet BM in the most populated 15 counties in Washington (blue) and California (orange) in 2050. The box represents the interquartile range (25th–75th percentiles) of county-level adoption rates of SocNet BM, the center line marks the median, and the whiskers indicate the minimum and maximum values. Cross markers show the adoption rate of State BM, star markers show the adoption rate of County BM, and square markers show the empirical data.

adoption diffusion, we calculate Moran's *I* statistics[47] for State/County BM, and SocNet BM based on their output adoption rate for each census tract (see the section "Fitting procedure"). Moran's I statistic is a commonly used spatial autocorrelation coefficient that is generally bounded between −1 and 1[48]. A positive Moran's I statistic (or a positive spatial autocorrelation) indicates that similar adoption rates are found in census tracts that are near each other; while a negative one suggests dissimilar adoption rates are found in tracts near each other.

With networks with $\gamma = 1, 3, 10$, calibrated models show diverse adoption predictions in spatial characteristics. Figure 5A shows the adoption rates with corresponding Moran's *I* statistics under $\gamma = 1, 3, 10$ in Los Angeles. When $\gamma = 1$, Moran's *I* statistics first increase and then decreases with growing adoption rates. In contrast, with $\gamma = 3$ and 10, Moran's *I* statistics monotonically increase with growing adoption rates. Figure 5B shows that Moran's *I* statistics of State/County BM are much higher than the data. For State/County BM, we only know the state/county-level aggregated adoption rate. To obtain census tract-level adoption rates, we assign adopters proportionally to the population in each census tract (see the section "Benchmark BM"). In summary, Fig. 5A and B demonstrate how different network structures influence the adoption by changing the $\gamma$ of networks in the SocNet BM. As different social network structures lead to different spatial patterns of adoption, we can infer the best social network model to represent the adoption based on the spatial auto-correlation of the data. Specifically, in Los Angeles, $\gamma = 10$ appears to better characterize the social network associated with PEV adoption (see Supplementary Information Note S6 for results in other counties). Overall, the SocNet BM with a spatial social network allows more flexibility to capture the spatial autocorrelation of the empirical diffusion. Figure 5C–E further show the geographical distribution of Los Angeles' adopters in 2050

under $\gamma = 1, 3, 10$. We find that with increasing $\gamma$, the number of adopters decreases (especially from the low-income group). The diffusion with $\gamma = 3, 10$ is slowed down mainly because a higher $\gamma$ reduces social network effects on the low-income group. Specifically, when the diffusion starts, it is driven by innovation-related factors, so high-income people adopt first. With higher $\gamma$, the high-income group is less likely to link with the low-income group, who usually live far away from them. Therefore, the adoption rate among the low-income group, whose adoption decisions are driven by social network effects, declines.

At last, we show that for SocNet BM, varying $p$ and $q$ can effectively fit the empirical data but show different predictions. Figure 6A–C shows that for the three income groups, high $p$ and low $q$, as well as high $q$ and low $p$, are all consistent with the data. However, these different combinations of $p$ and low $q$ could show remarkably different predictions. Figure 6D shows the Los Angeles adoption rate growth curve of varying $p$ and $q$ values, we can observe that all calibrations fit the data well. However, their predictions on the adoption rate in 2050 varied between 10% and 40%.

## Promoting PEVs adoption under uncertainty
In this section, we aim to explore campaign strategies to promote PEV adoption through innovation-related approaches and imitation-related approaches, with consideration of the uncertainty in modeling. We first propose an influencer strategy to promote PEV adoption. The influencer strategy refers to the campaign of distributing PEVs among influencers (agents with more connections in the social network) or asking them to recommend PEVs. Examples include sending PEVs or offering free leases of PEVs to influencers for a certain period, so as to let them share driving experience or the advantages of PEVs to

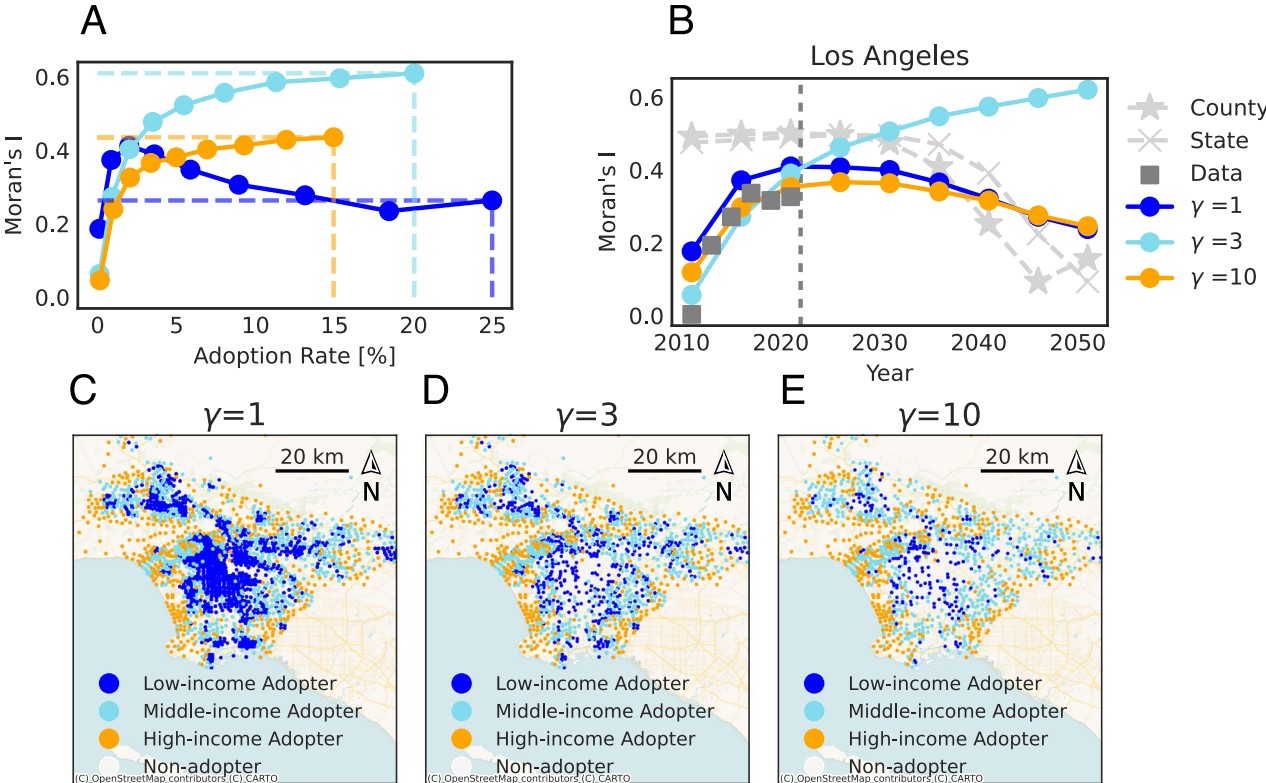

**Fig. 5 | Comparison of State/County BM and SocNet BM's geographical pattern in prediction.** State/County BM is based on a uniquely estimated parameter set, while SocNet BM results are obtained by averaging 10 simulations for each of four independently estimated parameter sets in Los Angeles. **A** Adoption rate with corresponding Moran's I statistics under $\gamma = 1, 3, 10$. Solid lines with circle markers with dark blue, light blue, and orange represent the median (50th percentile) Moran's I statistics and the corresponding adoption rate under the network with $\gamma = 1, 3, 10$, respectively. Dashed lines to the axis denote the median Moran's I statistics and the corresponding adoption rate in the year 2050. **B** Moran's I statistics for State/County BM and SocNet BM. For SocNet BM, we use the network with three $\gamma$s, i.e., 1, 3, 10. The fitting process is repeated for each $\gamma$, and the median of predictions is reported. Solid lines with circle markers show the median (50th percentile) Moran's I statistics from SocNet BM. Dashed lines with cross markers and star markers show Moran's I statistics from State/County BM; Squares show the Moran's I statistics calculated from empirical data on the tract level. **C**–**E** Adopters' geographical distribution in Los Angeles in 2050 under $\gamma = 1, 3, 10$. Dark blue/light blue/orange points represent adopters from the low-/middle-/high-income group, respectively.

their followers. In this strategy, we choose influencers based on their target audience. The target audience is defined as the income group from which the influencer has the most connections. Here we have four strategies: choosing influencers targeting low-income groups (Low), middle-income groups (Middle), and high-income groups (High), or randomly choosing people from the whole population as influencers, ignoring their connectivity (Random). For Low, Middle, and High strategies, we sort influencers by the number of connections to a target income group, then pick the top 100 and make them PEV adopters/advocates in 2023. We estimate the adoption rates with the four strategies in 2050.

Figure 7A shows the average adoption improvement of four strategies for various sets of $p$ and $q$ parameters calibrated from empirical data. The adoption improvement is calculated as the difference in adoption rates before and after the campaign in Los Angeles in 2050, divided by the adoption rates before the campaign. We find that for networks with more distant connections ($\gamma = 1$), strategies targeting different socio-demographic groups (Low, Middle, High) have similar effects, which are all much better than the Random strategy. The similarity of Low, Middle, and High strategies is because (1) people who live close are more likely to have similar income levels, and (2) lower $\gamma$ leads to more distant links. Therefore, people are more likely to have connections with different income groups when $\gamma = 1$. This leads to the situation that those influencers who have more connections with one group also have more connections with the other two groups. As $\gamma$ keeps increasing, the network has fewer distant links

and a higher level of spatial clustering, and thus influencers are more likely to have higher effects on one of three groups. Interestingly, we find that the adoption improvement is smaller with increasing $\gamma$. That means that if people in the region of interest have more distant connections, the influencer strategy is more efficient. In contrast, if people tend to connect with only nearby contacts, the influencer strategy is less efficient.

Figure 7B–D compares innovation and imitation-related adoption boosting strategies, applied to the low-income group and for three pairs of $\hat{p}_{low}$ and $\hat{q}_{low}$ that fit current data. We represent the "innovation-related" strategy as an increase of $\hat{p}_{low}$ in a fixed factor, which modifies the global propensity of innovation adoption within the low-income group. We represent the "imitation-based" strategy by setting the first $n$ agents with the highest degree as adopters. Each figure shows the years to critical adoption rates (16%) resulting from the innovation-related and the imitation-related strategies for each pair ($\hat{p}_{low}, \hat{q}_{low}$). Figure 7B illustrates that with a relatively low fitted $\hat{p}_{low}$ and high $\hat{q}_{low}$, an imitation-related approach with more influencers can effectively boost adoption, whereas increasing the fixed factor (innovation-related strategy) does not have much effect. Conversely, Fig. 7D depicts a scenario with a relatively high $\hat{p}_{low}$ and low $\hat{q}_{low}$, where increasing fixed factors substantially enhances adoption, but adding more influencers has a negligible effect. Figure 7C presents a middle ground, where combining both strategies achieves the highest adoption increase. It is crucial to note that the scenario where imitation-related strategies have a stronger impact does not imply that policies

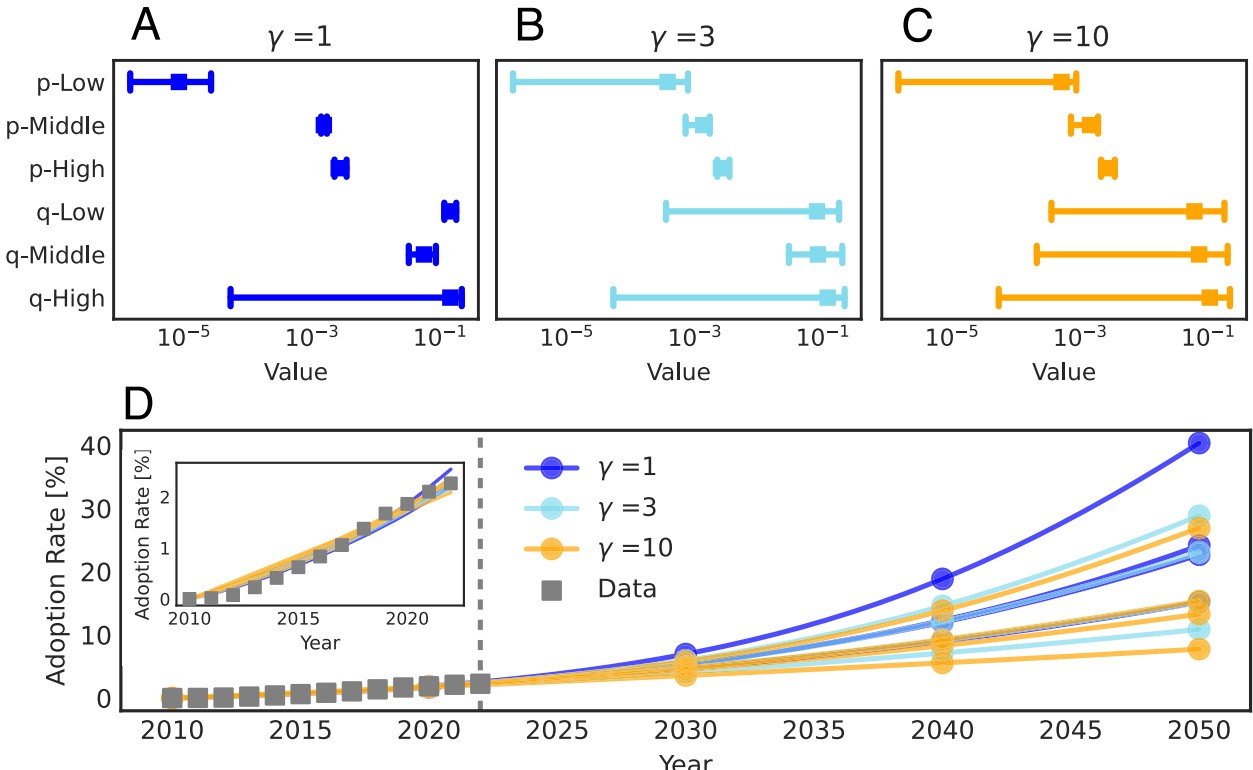

**Fig. 6 | Comparison of calibrated SocNet BM's *p* and *q* sets and corresponding adoption rate prediction.** Results are obtained by averaging 10 simulations for each of four independently estimated parameter sets in Los Angeles.
**A**–**C** Calibrated *p* and *q* values of low-/middle-/high- income group when using the network with three *γ*s, i.e., 1,3,10. Square markers show the mean value, and the error bar indicates the min–max interval of *p* and *q* values. **D** County-level adoption rate growth curves using calibrated *p* and *q* sets of SocNet BM. Each Line with circle markers shows the adoption rate curve of one calibrated *p* and *q* set. The color of the lines indicates the network structure (*γ*) used for SocNet BM calibration. Squares show the empirical adoption rate growth curve. The inset is the same plot but with a zoom-in period from 2010 to 2022.

and actions seeking a global effect do not matter. On the contrary, initiatives such as consumer incentives or the development of charging infrastructure are essential for facilitating adoption in low-income areas or regions with limited charging stations. However, which are the best actions to increase the innovator rate is not clear. For example, Slowik et al.[23] assess the PEV market in the 50 most populous metropolitan areas in the United States. They demonstrate that for areas with high PEV penetration, there is heterogeneity in the availability of policies, charging infrastructure, consumer incentives, and vehicle models. Altogether, Fig. 7B–D shows that with different $\hat{p}_{low}$ and $\hat{q}_{low}$ effectively fitting the data, the year to reach critical adoption for different combinations of imitation-related strategy and innovation-related strategies varies, as the contour of the year achieving critical adoption ranges from horizontally displayed to vertically displayed. In this regard, due to the small amount of data available, a conclusion on which combinations are the best is elusive. However, once we move beyond the early adoption stage (when adoption reaches the majority stage, and thus can obtain stable *p* and *q* from data), we can then identify whether an influencer strategy is appropriate, as suggested by Fig. 7B–D, as well as find the best target audience according to Fig. 7A by considering the network structure of the users.

## Discussion

Modeling and projecting PEV adoption helps us plan resources to create a viable PEV ecosystem. However, forecasting PEV adoption is challenging because it is at a very early stage, and even fitting simple models can be difficult. Today, a benchmark model for projecting adoption is the two-parameter BM. One important ingredient to be added is the role of spatial social networks in its forecast.

Our comparison of projections for future PEV adopters reveals that a more sophisticated diffusion model, fitted equally well to the same data, can produce results drastically different than the traditional BMs. The divergence of forecasts also indicates that it's important to consider the complexity of the current model design. While the more parameters in these models facilitate a closer fit to the available data, there is an inherent risk of overfitting. Furthermore, this study demonstrates that the SocNet BM with the spatial social network as an extra ingredient allows more flexibility to capture the spatial autocorrelation of the empirical diffusion process.

This work also explores PEV promotion campaigns' impact under uncertainty. We explore multiple possible social networks. Our analysis reveals that with more distant links in the social network, influencer strategies targeting different income groups yield similar adoption improvements by 2050. With more close links, the effectiveness of targeting specific income groups increases. Overall, the study emphasizes the need for adaptable, context-specific approaches that consider local socio-demographic contexts and social network structures when designing adoption-boosting strategies.

This study also provides implications for power system planning. Primarily, PEV adoption is uneven across regions. This framework provides a benchmark to predict the geographical distribution of PEV adopters. In areas with high PEV adoption, the stress on transmission and distribution networks intensifies. If the charging of these adopters remains uncoordinated, it may cause transformer overloading[49], overload transmission lines[50], and early replacement of equipment[51]. This requires strategic control and infrastructure build-out to avoid bottlenecks and maintain reliable service[11]. Additionally, given demographic disparities in PEV adoption, equitable planning for charging

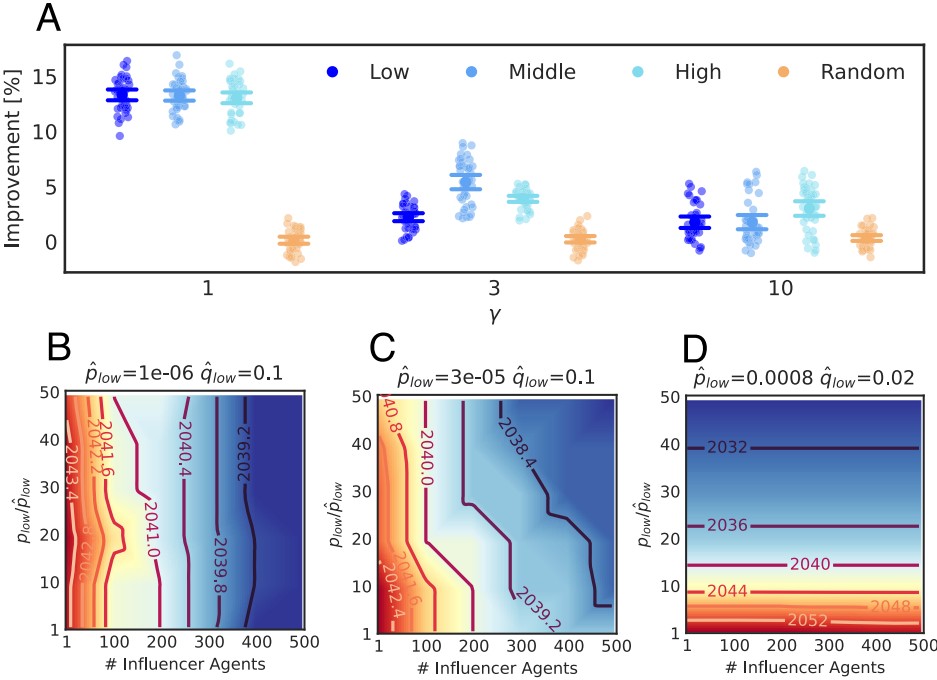

**Fig. 7 | Campaign to promote PEV adoption under uncertainty in space and time.** Results are obtained by averaging 10 simulations for each of four independently estimated parameter sets in Los Angeles. **A** Adoption improvement under different $\gamma$. For each $\gamma$, we show results for four campaign strategies: Using influencers targeting low-income group (Low), middle-income group (Middle), high-income group (High), or randomly picking influencers (Random). Points represent adoption improvement of 40 replicates under four independent fitting and simulation processes. The error bars show the mean ± two standard errors among all points. **B–D** Heatmap and contour of the year to reach critical adoption (16%) with different $p_{low}$ and the number of influencer agents. Here we show three cases of different $\hat{p}_{low}$ and $\hat{q}_{low}$. Three different $\hat{p}_{low}$ and $\hat{q}_{low}$ are obtained from fitting data.

**Table 1 | Socio-demographic characteristics of the three groups**

|  | Low-income Group | Middle-income Group | High-income Group |
|---|---|---|---|
| Washington income interval [US$] | <66,685 | 66,685–92,830 | >92,830 |
| California income interval [US$] | <64,623 | 64,623–95,587 | >95,587 |

infrastructure is crucial. The results show that middle- and low-income groups are less inclined to adopt PEVs due to innovation-related factors compared to the high-income group. To encourage PEV adoption among these communities, grid operators may need to offer lower electricity rates or enhance grid capacity to support new charging stations nearby. Moreover, the uncertainty in adoption forecasts demands integrating distributed energy resources, such as energy storage[52], renewables[53], vehicle-to-grid solutions[54], to bolster grid resilience.

This research may be expanded in various ways. For instance, integrating technology, policy, and built-environment evolution into diffusion predictions could be one area of focus. Alternatively, planning for power and transportation infrastructure in response to varying forecasts of PEV adoption represents another avenue. Lastly, this study uses data prior to 2023, but we note that the BEV stock in California was 3.8% by the end of 2023[55], which crosses from the innovator phase into the early adopter phase. Larger numbers of adopters could provide new opportunities to predict the diffusion in a more detailed way. For example, while segmentation based on consumer characteristics alone is a valid method for forecasting PEV adoption[56], the consideration of the vehicle models is crucial to connect adoption with the power grid. Currently, due to limited data, we focus on PEVs without distinguishing between BEVs and PHEVs. With more data, it could become feasible to predict the adoption of BEVs and PHEVs separately and to investigate the distinct implications for the power grid related to BEVs versus PHEVs, along with consumer profiles and their motivation.

## Methods

### Market segmentation
Previous research about understanding the driving forces behind PEV adopters' decisions highlights that adoption choices are related to household incomes[16,17,39]. Therefore, we divide the potential market into three socio-demographic groups: low-income group (income below 33% of the reference population of the state), middle-income group (income between 33% and 66% of the reference population of the state), and high-income group (income above 66% of the reference population of the state). The socio-demographic characteristics of the three groups are shown in Table 1. We justify the choice of this segmentation approach in Supplementary Information (see Supplementary Information Note S7).

### Benchmark BM
We first revisit the BM and then illustrate the State/County BM and its fitting procedure.

**Revisit BM.** The Bass diffusion model enables us to investigate adoption dynamics[45]. This can be done by fitting the cumulative distribution function of the adoption data with the cumulative distribution function of the model. The Bass cumulative distribution function is defined by

$$\frac{dy(t)}{dt} = (p + q/m \times y(t))(m - y(t)),\qquad(1)$$

where $y(t)$ is the number of adopters at time $t$, $p$ and $q$ are the parameters of adoption, and $m$ is the market size. This nonlinear differential equation can be solved by

$$y(t) = m \frac{1 - e^{-(p+q)t}}{1 + \frac{q}{p} e^{-(p+q)t}}. \qquad (2)$$

**State/County BM.** The State/County BM fits empirical state-aggregated data for three socio-demographic groups. Specifically, For customers belonging to group $i$ in state (county) $j$, the market size $m_j^i$ is the number of vehicles of group $i$ in state (county) $j$, and $t = 2010, ..., 2022$. We obtain $(p_j^i, q_j^i)$ where $i \in \{$Low, Middle, High$\}$ and $j \in \{$Washington, California$\}$ (or $j \in \{$Los Angeles, ..., King$\}$) by fitting Eq. (2) through least-squares method[57].

**Assignment of state/county adopters to tracts.** To obtain census tract-level adoption rates from State/County BM, we assigned aggregated adopters proportionally to the population in each census tract. Denote the number of adopters of group $g$ in state (county) $l$ as $y_l^g(t)$, and the number of vehicles of group $g$ in state (county) $l$ as $m_l^g$. Then, for the tract $h$'s group $g$ with $m_h^g$ vehicles, the number of adopters in group $g$ is calculated as

$$y_h^g(t) = y_l^g(t) \times \frac{m_h^g}{m_l^g} \qquad (3)$$

## SocNet BM

In this section, we first explain how we construct the social networks. Then we present the SocNet BM's simulation and fitting procedure.

**Social network construction.** We construct social networks with ingredients of preferential attachment and distance selection by following[46]. For the region of interest, we sample nodes from data with expansion factor = 5 by keeping spatial distribution (an expansion factor is defined as the empirical number of vehicles from census/ number of nodes). The sensitivity analysis for various expansion factors is available in the Supplementary Information (see Supplementary Information Note S8). These nodes are in a two-dimensional space described by the longitude and latitude of potential adopters' home locations (the geographical resolution is at the census tract level). Then we can calculate the distance between any adopter $i$ and adopter $j$ as $d_{ij}$. Once the nodes are distributed in this space, we construct links using the following algorithm: (1) Select at random a subset of $n_0$ initial active nodes. (2) Take an inactive node $i$ at random and connect it with an active node $j$ with probability:

$$\text{prob}_{i \to j} \propto \frac{K(k_j)}{D(d_{ij})}, \qquad (4)$$

where $k_j$ is the connectivity of node $j$, $d_{ij}$ is the distance in kilometers between nodes $i$ and $j$, $K(.)$ and $D(.)$ are given functions, and $\text{prob}_{i \to j}$ is then normalized up to a factor $\sum_{j \in \text{active nodes}} \text{prob}_{i \to j}$. For each inactive node, we sample $k_r = 3$ edges such that the average connectivity is $\langle k \rangle = 2k_r$, and then mark the node $i$ active. Finally (3), repeat (2) until all nodes are active. To generate preferential attachment and empirical spatial features, we use $K(k) = k + 1$ and $D(d) = d^\gamma$, where $\gamma = 1, 3, 10$ (see Supplementary Information Note S9 for the rationale behind the selection of $\gamma$ values). We illustrate the algorithm with an example in the Supplementary Information (see Supplementary Information Note S10).

**Simulation procedure.** This model consists of $n$ agents indexed by $i \in \{1, ..., n\}$, each of which is in either of two states: potential adopter or adopter. We use a set of variables $v(t)$ to describe the agents' adoption state (i.e., $v_i(t) = 1$ when agent $i$ has adopted at $t$). Analogously to the

BM, the probability of agent $i$ to adopt depends linearly on an independent parameter $p$ and a parameter $q$ that depends on the fraction of prior adopted neighbors. Agent $i$'s probability to transition from non-adopter to adopter at $t$ is as follows:

$$f_i(t) = (p + \frac{\sum_{j \in z_i} v_j(t)}{|z_i|} q)(1 - v_i(t)), \qquad (5)$$

where $z_i$ is the set of agents having connections with node $i$. This formulation implies that each agent's probability of adoption is influenced uniformly by the adoption state of all other neighbor agents.

A synchronous updating formulation of the BM is achieved by $\bar{v}_i(t)$ which is used to store the new state of the system until the end of the period when the actual updating occurs[58]. In each time $t \in \{2010, ..., T\}$, the algorithm decides for each agent $i$ whether or not it adopts based on the adoption probability according to Eq. (5) and a random value drawn from a uniform distribution $U(0, 1)$. If an agent adopts, the temporary variable $\bar{v}_i(t)$ is updated accordingly to 0 or 1. As soon as all agents have made their adoption decisions, the state of the system is updated as $v_i(t) = \bar{v}_i(t)$ for $i \in \{1, ..., n\}$.

**Fitting procedure.** We follow the method in ref. 59. Without loss of generality, we take one county as an example to illustrate the procedure. The procedure consists of two stages: locating the initial point and searching for the optimal solution.

**Locating the initial point.** In this stage, we aim to determine the explicit connection between the BM and a particular SocNet BM in terms of parameters. In detail, for a given county, we generate adoption curves with all possible parameter pairs $(p_{\text{SocNet}}, q_{\text{SocNet}})$ using SocNet BM, where $p^{\text{Low}}, p^{\text{Middle}}, p^{\text{High}} \in \{1 \times 10^{-4}, 5.5 \times 10^{-4}, 1 \times 10^{-3}\}$ and $q^{\text{Low}}, q^{\text{Middle}}, q^{\text{High}} \in \{1 \times 10^{-3}, 5.5 \times 10^{-3}, 1 \times 10^{-2}\}$. Then we fit Eq. (2) to all generated adoption curves with the nonlinear least squares method. Afterwards, we get estimated $(p_{\text{Bass}}, q_{\text{Bass}})$ pairs corresponding to the $(p_{\text{SocNet}}, q_{\text{SocNet}})$. We then estimate $C$ and $\epsilon$ in the following linear model by ordinary least squares linear regression[60] to represent their relationship

$$\begin{bmatrix} p_{\text{SocNet}} \\ q_{\text{SocNet}} \end{bmatrix} = C \cdot \begin{bmatrix} p_{\text{Bass}} \\ q_{\text{Bass}} \end{bmatrix} + \epsilon. \qquad (6)$$

With this relationship, we fit the BM to the yearly sales data of PEVs from 2010 to 2022 in Washington and California using again the nonlinear least-squares method. From this fit, we get $(\hat{p}_{\text{Bass}}, \hat{q}_{\text{Bass}})$. Substituting these values into Eq. (6), we get initial points as $(\hat{p}_{\text{SocNet}}, \hat{q}_{\text{SocNet}})$.

**Searching for the optimal solution.** In this stage, we search for the optimal parameters around initial points using the Luus–Jaakola heuristic[61]. Starting out from $(\hat{p}_{\text{SocNet}}, \hat{q}_{\text{SocNet}})$ pair, we set up a grid search space as $\mathcal{X}$:

$$\mathcal{X} = \prod_{g=1}^{G} (\mathcal{X}_p^g \times \mathcal{X}_q^g), \qquad (7)$$

where $\mathcal{X}_p^g$ and $\mathcal{X}_q^g$ are search space for $p$ and $q$ of for group $g$, respectively. Denote the search interval as $\Delta = 0.9$, search center as $\hat{p}_{\text{SocNet}}^g$ and $\hat{p}_{\text{SocNet}}^q$, we have

$$\mathcal{X}_p^g = \hat{p}_{\text{SocNet}}^g \times \{1 - \Delta, 1, 1 + \Delta\}, \qquad (8)$$

$$\mathcal{X}_q^g = \hat{q}_{\text{SocNet}}^g \times \{1 - \Delta, 1, 1 + \Delta\}. \qquad (9)$$

Then we run SocNet BM corresponding to the randomly sampled subsets $\boldsymbol{x}_i \in \mathcal{X}, i \in \{1, ..., 50\}$ drawn from $\mathcal{X}$, and calculate the

goodness of fit for $\boldsymbol{x}_i$ as

$$s(\boldsymbol{x}_i) = \max_{g=1,\,\ldots,\,G} \frac{1}{T} \left| \sum_{t=1,\,\ldots,\,T} (y^g(t,\boldsymbol{x}_i) - Y^g(t))\right|^2, \quad (10)$$

where $y^g(t, \boldsymbol{x}_i)$ is simulated number of adopters with parameters $\boldsymbol{x}_i$ at time $t$, and $Y^g(t)$ is the empirical number of adopters for group $g$ at time $t$. Until the maximum number of 100 iterations is performed or the fitness is unchanged for 50 iterations, repeat the following: Denote optimal solution so far as $\boldsymbol{x}^*$ with goodness of fit $s(\boldsymbol{x}^*)$. If $s(\boldsymbol{x}_i) < s(\boldsymbol{x}^*)$, then update searching center points as $\boldsymbol{x}^* = \boldsymbol{x}_i$ and set $\Delta = 0.9$. Otherwise, set $\Delta = \Delta \times r$, where $r = 0.9$ is the searching decay rate, and update $\mathcal{X}$ with $(\hat{\boldsymbol{p}}_{\text{SocNet}}, \hat{\boldsymbol{q}}_{\text{SocNet}}) = \boldsymbol{x}^*$ and $\Delta$.

To capture the uncertainty in the fitting process, we repeat the fitting procedure five times for each research area.

### Measuring of spatial autocorrelation of adoption

To measure spatial autocorrelation of the diffusion process, we calculate Moran's $I$ statistics for each given time slot $t$ based on:

$$I(t) = \frac{N}{\sum_{i=1}^{N}\sum_{j=1}^{N} w_{ij}} \frac{\sum_{i=1}^{N}\sum_{j=1}^{N} w_{ij}(F_i(t) - \bar{F}(t))(F_j(t) - \bar{F}(t))}{\sum_{i=1}^{N}(F_i(t) - \bar{F}(t))^2}, \quad (11)$$

where $N$ denotes the number of census tracts; $w_{ij}$ denotes the Euclidean distance between the centroid of census tract $i$ and census tract $j$; $F_i(t)$ denotes the adoption rate of census tract $i$ at time $t$; $\bar{F}(t)$ is the mean adoption rate of all census tracts at time $t$.

### Reporting summary

Further information on research design is available in the Nature Portfolio Reporting Summary linked to this article.

## Data availability

The data supporting this study are available at https://doi.org/10.5281/zenodo.17190814.

## Code availability

The code used in this study is available at: https://zenodo.org/records/17199127.

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

## Acknowledgements

M.G.C. was supported by the ITS-SB1 Berkeley Statewide Transportation Research Program and the California Air Resources Board.

## Author contributions

J.W., A.S., and M.C.G. conceived the research. J.W., A.S., and M.C.G. developed the methodology. J.W. processed the data, implemented the methodology and performed visualization. J.W., A.S., and M.C.G. analyzed the results. J.W. prepared the original draft. J.W., A.S., and M.C.G. edited and revised the manuscript. M.C.G. supervised the research.

## Competing interests

The authors declare no competing interests.
