## [Transparent Peer Review file · Nature Communications]

Planning the Electric Vehicle Transition by Integrating Spatial Information and Social Networks

Corresponding Author: Professor Marta González

Version 0:

Reviewer comments:

Reviewer #1

(Remarks to the Author)

Key results:

Authors compare the results of a Bass diffusion model which does and doesn't control for spatial social networks to forecast EV adoption. The model which takes into account spatial effects yields adoption rates much lower than the one without. Additionally, authors emphasize the relevance of their results for more effective, targeted marketing messages.

Validity:

the results are interpreted in a valid way. However, it was unclear after reading the manuscript where they obtained data for incentive knowledge, or why they chose this particular variable, with appropriate literature showing its significance.

In addition, the following statement could be discussed in further detail:

"Altogether, this means that when the population is more sensitive to intrinsic factors, establishing new financial incentives for the whole potential market, increasing charging station availability, and reducing the upfront cost of adopting EVs will be more efficient. Otherwise, influencers recommend EVs to their friends, or sending EVs to influencers will be more efficient."

Specifically, there is literature which explores the different effects of charging infrastructure and financial incentives, with some finding that funding would be more effective if spent on charging infrastructure compared with incentives, and others finding that investments in home charging would be more effective than public charging. There is also a white paper by the ICCT titled "EXPANDING THE ELECTRIC VEHICLE MARKET IN U.S. CITIES" by Slowik et al. which explores the availability of incentives and infrastructure across U.S. cities and finds heterogeneity in cities with high PEV penetration and availability of incentives and charging.

Additionally, authors could expand to clarify cases where lower-income households are more motivated intrinsically or extrinsically. It mentions above that in LA, influencer targeting is more effective for lower-income households, so would that mean they are more extrinsically motivated, and thus need less incentives and charging?

Significance:

The significant results indicate that social networks with closer connections lead to more clustered adoption, compared with more distant connections, which leads to more balanced diffusion. Additionally, targeting specific income groups is most effective for close connections, whereas "influencer" strategies are most effective for social networks with longer distance connections.

Data and methods:

the data and presentation of data are valid.

Analytical approach:

Could be defended more robustly, i.e. see comment in suggested improvements regarding mentioning other modeling alternatives.

Suggested improvements:

I was surprised that household income is the only demographic used to segment, considering the prominence of adopters living in/owning a detached home. I think this dimension is worth mentioning at the very least.

Clarity and context:

It may be helpful to highlight the benefits of the socNet BM more in the introduction by incorporating more information that is presented in the first paragraph of section 2.2.

Introduction could clarify what is meant by "electric vehicle". i.e., are authors combining BEV and PHEVs? surely there would be different power grid implications for BEVs vs. PHEVs, in addition to consumer profile and motivation.

"intrinsic" and "extrinsic" could be more clearly defined, especially if it is being used in the introduction, and are the central components of the Bass model. Extrinsic also includes network effects, word of mouth, and marketing, not just social media interactions, as is currently stated in the intro, correct? Lee et al. (source 9) considers marketing to be an intrinsic factor. Perhaps more justification is needed on selection of factors, and perhaps being more explicit that these are the components of the BM.

Clarity on how knowledge of incentive policies was measured would be appreciated-- financial incentives only? It would also be useful to justify the use of this variable, as to how it would differ from perhaps availability of incentives? or take into account income caps?

Context of what previous studies which use Bass model have found could improve the introduction. In addition to providing context of other method approaches, such as choice modeling, or regression models of disaggregate or aggregate data, for example, in addition to the Bayesian approach that is currently mentioned.

Introduction could overall be improved by providing context of various mandates of EV sales, which further motivates the need to model the effects on power grid, in addition to being more specific about the socio demographic profile of early PEV adopters.

There seems to be a disconnect between the introduction emphasizing implications for the power grid, and the discussion not mentioning this at all.

References:

Reference 7 is about fleet decisions, not sure if it's appropriate since the scope of study is on consumer decisions? Further, on the same line, I believe "choices of early adopters" could be improved to be more specific.

References 9,10, & 11 could be specific about what kinds of demographics are related to BEV adoption. For example, source 11 is about home charge access; additionally, it may be appropriate to differentiate between home and public charging access (reference 14).

I do not believe that references 12 & 13 are appropriate citations for "knowledge of incentive policy". Reference 12 measures the effect of financial incentives for PEV adopters, and reference 13 is about the effect of various policy scenarios on business strategy; neither are within the scope of measuring consumer knowledge of incentives. To that end, I'm not sure reference 13 is appropriate for a paper focusing on consumer decisions.

Regarding extrinsic motivation (reference 15 and 16), would not in-person interactions also be considered extrinsic? Perhaps more references are needed for word-of-mouth, or workplace network effects.

I do not believe references 32 and 33 are appropriate for the statement that "adoption choices are related to household income" 32 is regarding the business case and supply chain for PEVs, the other is mostly about being "eco-friendly", with income not being mentioned in the abstract.

References could overall be improved by more robustly choosing appropriate literature for each component of the BMs, and incorporating the literature into the discussion, as there are currently no references in the discussion.

My expertise:

I do not have experience running my own Bass diffusion model, therefore most of my feedback is to highlight the benefits of using such an approach, to someone who is less familiar with the approach.

Kelly Hoogland

(Remarks on code availability)

Reviewer #2

(Remarks to the Author)

This study proposes an innovative approach that integrates spatial information and social network into a technology diffusion model to predict long-term EV adoption. The findings are significantly different from existing studies, showing a

much more conservative adoption rate. This finding emphasizes the importance of including spatial and social network structures in EV technology diffusion and adoption prediction. The paper is well-written and well-structured. However, several comments need to be addressed before publishing.

1. One significant limitation of the paper is the lack of consideration of the possibility that EV users might switch back to gasoline vehicles. This study assumes that once consumers adopt EVs, they will not switch back, overlooking the behavioral uncertainty in the market. Many factors such as price changes in fuel/electricity, charging infrastructure availability, and shifts in consumer preferences could lead to a technology reverse migration. Report shows that about 46% of current EV users in U.S. are likely to switch back to ICE[1]. Such impact has been included in the data (e.g., number of EV titles) but the model ignored. The omission of this reverse migration could lead to an overestimation of EV adoption. This possibility should be incorporated into the modeling framework to provide a more comprehensive forecast of EV adoption. For example, in the simulation process, each agent that adopted EV should be assigned a probability of abandoning EV after several years of usage.

2. For the method part, the reviewer has one concern in Section 4.3.1, where the authors arbitrarily choose $n=50,000$ nodes to construct the social network in two different cities. However, it lacks a clear rationale for this selection. It also raises concerns about the robustness of the results. First, the paper does not provide any rationale for why this specific number of nodes was chosen, nor does it discuss how this choice might affect the model's predictions. The number of nodes in a social network model is a critical parameter that may significantly affect the simulation process. Second, different cities have varying population sizes, social structures, and urban layouts. All of them require different network sizes to capture the heterogeneity. Assigning 50,000 nodes for different regions of interest oversimplifies the social structures. For example, a city with a larger population or more complex human interactions might require a denser or larger network to represent the social structure. When stakeholders from other regions/cities want to apply the model from this study, it is not clear how to choose the appropriate node count for a better prediction. Third, without a sensitivity analysis, it is unclear whether the model's results are reliable or whether they could change significantly with a different number of nodes. This uncertainty weakens the model's prediction performance and limits its applicability to real-world scenarios. Last, it is unclear whether/how real world location data is used in the social network construction. Because SocNet BM is the major contribution of this work, it could benefit from having more technical details (e.g., in the Supplementary Information) to improve method clarity.

3. Figure 5 (C,D,E) indicates that the urban layout could be a very important factor. It directly affects both the spatial and social structures of a city/county. Different regions/cities have varying urban designs, including different grid sizes and population densities. For example, densely packed urban areas could have quicker adoption because of strong geographical and social connections, while sprawling suburban areas may show slower diffusion due to greater distances between adopters. The urban layout is highly correlated with spatial and social structure, which further can affect EV adoption. The reviewer expects to see more analysis and discussion about the correlation between urban layout and adoption diffusion patterns. Understanding this correlation can provide more insights into future EV adoption dynamics.

4. When evaluating the adoption rate error for 2022, which years of data were used to fit the model? Also, it is unclear whether the "adoption rate error" values are absolute values in terms of adoption percentage or percentage values relative to the ground truth data. Please clarify. It would also be helpful to show the overall performance of each model for the entire states, considering that some models may overperform the others in some counties but underperform in the others.

Below are some additional minor comments:

1. The code link is not valid. Please double-check.
2. Page 4. Most of the content in the paragraph "Fig. 2 (A, C) show ..." has been included in the figure caption. The figure caption is informative enough to understand the figure independently. In addition, it is not clear the meaning of the filled area between two curves in Fig. 2 (A, C). More explanation should be included.
3. In Figure 2 (E, F), the data points overlap with the box, making it hard to compare the model fitness of different counties. Separating the box and data points into two columns for each county can improve the readability of this figure.
4. The explanation of parameters p and q in the first paragraph of section 2.3 should be integrated with the similar content in section 2.2.
5. Typo: "Our comparison of projections for future EV adopters reveals that by introducing social network, the number of adopters by 20250 is 25% of benchmark", which should be "2050".
6. Figures in the Supplementary Information could benefit from better caption and legend formatting.

Reference

[1]. McKinsey & Company. McKinsey Mobility Consumer Pulse. https://executivedigest.sapo.pt/wp-content/uploads/2024/06/Mobility-Consumer-Pulse-2024_Overview.pdf (2024).

(Remarks on code availability)

I attempted to review the code but the link doesn't work.

Reviewer #3

(Remarks to the Author)

Title: Planning the Electric Vehicle Transition by Integrating Spatial Information and Social Networks

Reviewer's note: Thank you to the authors for submitting this paper. I enjoyed reading it and I think it is an important contribution to the literature. I hope that the below comments will help the authors in making the paper as impactful as possible.

Summary:

The paper presented concerns an alternate approach to modeling BEV adoption using diffusion of innovations theory. The specific advance is the use of an augmented BDM which accounts for the social network among the population of potential adopters. The paper highlights a stark contrast between long term predictions generated using the SocNet BDM and traditional BDM using the same data. Challenging the use of the BDM for EV adoption is very relevant as so much policy is based on it and it is probably too simple to be useful. This result is important and fitting for Nature.

The paper itself requires changes. In particular, the authors make many claims about the physical meaning of model parameters which are not justifiable. The parameters of the BDM are descriptive and fit to data - the authors do not provide or cite proof that the p -parameter is correlated to vehicle net cost, income, etc. but this assumption is manifested throughout the paper. The authors do not sufficiently justify that the SocNet model is a better reflection of reality than a geometry based model but do claim this several times. The SocNet model introduces an additional parameter that must be either assumed or fit to data - the authors test three values for this parameter but do not justify why they chose these values.

That the SocNet BDM can produce such different values for the same input data is an interesting finding. However, in both the introduction and discussion the authors frame this result as the BDM majorly overpredicting the more realistic SocNet BDM. The problem is that the SocNet BDM is not obviously proven to be more realistic and the SocNet BDM produces a wide range of predictions depending on the gamma parameter. As written, this paper proves that a different set of assumptions than usually applied leads to very different conclusions, not that these conclusions are more accurate.

The repository either does not exist or is private.

Notes:

1. "Interestingly, while both models fit current adoption data until 2022, when projecting into the future, models that omit spatial social networks will result in four times more adopters by 2050." Could be reworded to be more direct, less biasing, and even emphasized as this is the key finding. Suggestion: "Pre-2022 EV adoption has largely been driven by personal decisions rather than social interactions, a dynamic which cannot continue. Thus both models can be fit to pre-2022 data but the decision to include or omit the effects of social networks results in a factor of four divergence in predictions by 2050." or similar.
2. This study uses 2022 and prior year data which is fine but the authors should note where there has been significant development by time of submission. Per CalTrans the BEV stock in California was 3.8% by the end of 2023 (<https://www.energy.ca.gov/data-reports/energy-almanac/zero-emission-vehicle-and-infrastructure-statistics-collection/light>) which crosses over into Early Adopter territory.
3. Figure 1: C, F - confidence intervals are barely visible and trend line is invisible where observations are thick. It would be better if the authors could somehow zoom in on California, perhaps by showing a "zoom-in" on the bay area.
4. Figure 2: B, D - The legend is cut off by the bars in both cases.
5. General comment is that the size and color scheme adopted in Figures 1 and 2 does not help legibility. The authors could choose colors that stand out better against a white background, change to a darker background, increase figure size, add grid lines, etc.
6. Table 1 and two paragraphs below: The authors make a great deal of interpreting fitted BDM coefficients. Specifically they give the reader the impression that these coefficients are fundamental behavioral characteristics of the relevant populations. They are not - they are descriptive. Specifically, the authors do not prove that a higher p -coefficient is due to "higher household incomes, they know more financial incentives policies, or they have more access to charging stations". Please either provide evidence to back this claim or remove it.
7. Second Paragraph of 2.4. The authors express an opinion here (and in the introduction) that the SocNet model has a more realistic interpretation of interaction and influence but insufficient evidence is provided to justify this claim. Specifically the authors argue "people in a more realistic social network are only influenced by their social contacts". Without contrary evidence, it is not ridiculous to think that consumers may be influenced by seeing EVs around even if they don't know the owners. With the evidence presented the authors must state that the SocNet model is based on different assumptions without claiming that these are more realistic.
8. Section 2.5: Insight into a physical manifestation of the effects of gamma parameter would be useful. For example - could the authors plot distributions of valency and edge length for different gamma? - Could the authors explain the selection of 1, 3, 10 and which they think is most realistic/why?
9. Section 2.5: There is a lot of information in this section but the most important point is that spatial autocorrelation seems to line up best with the gamma = 1 data. The authors could both emphasize this and its implications and reduce the rest of the discussion.
10. Figure 5: Panel A - what data did the authors use to compute the I statistic in the series labeled "data" (gray squares)? Is this tract level?
11. Section 2.6: This section is technically sound but all previous comments on the link between the model and reality are relevant. Also what do the authors think the "influencer" strategy would entail? Would a company distribute personalized incentives to individuals based on social media following?
12. Discussion: 20250 should be 2050

13. Discussion: The authors should consider what the key takeaways of this study are. In the opinion of the reviewer the strongest points are that (1) A more sophisticated diffusion model, fitted equally well to the same data, can produce results drastically different than traditional BDM implying substantial risk (2) The SocNet BDM with low gamma best reflects the observed spatial autocorrelation of BEV adoption.

14. Methods: Method descriptions are sound and well presented.

(Remarks on code availability)

The code is not publicly available at the address provided. I copied the URL and got an error 404. I went to the HuMNet lab git page and did not find it among the public repositories. Thus I did not review the code.

Reviewer #4

(Remarks to the Author)

(Remarks on code availability)

The code link is not valid

Version 1:

Reviewer comments:

Reviewer #2

(Remarks to the Author)

The authors have done a great job to address reviewer comments. We do not have further concerns and support accepting this manuscript.

(Remarks on code availability)

Reviewer #3

(Remarks to the Author)

All of my comments have been well addressed, thanks!

I recommend publication.

(Remarks on code availability)

The code is servicable for validating results but is not in the format of a generally usable package. The code is largely not commented but descriptive variable names make it readable. Nevertheless, I would instruct a student to use the code as inspiration rather than attempt use the code itself.

It would be nice if this was a PyPI package or, at least, structured like one. This would include the dependency management piece so users wouldn't have to set up an env manually.

Reviewer #4

(Remarks to the Author)

(Remarks on code availability)

Paper results are reproducible based on the published code.

Response to Reviewers:
“Planning the Electric Vehicle Transition by Integrating Spatial
Information and Social Networks”

Jiaman Wu, Ariel Salgado and Marta C. Gonzalez

February 28, 2025

Comments from reviewers in bold text — Responses to reviewers in plain text
~~Deleted text~~ — Added text

Comments from Reviewer 1

Authors compare the results of a Bass diffusion model which does and doesn't control for spatial social networks to forecast EV adoption. The model that takes into account spatial effects yields adoption rates much lower than the one without. Additionally, the authors emphasize the relevance of their results for more effective, targeted marketing messages. The significant results indicate that social networks with closer connections lead to more clustered adoption, compared with more distant connections, which leads to more balanced diffusion. Additionally, targeting specific income groups is most effective for close connections, whereas "influencer" strategies are most effective for social networks with longer-distance connections. The data and presentation of data are valid. The analytical approach could be defended more robustly, i.e. see the comment in suggested improvements regarding mentioning other modeling alternatives. I do not have experience running my own Bass diffusion model, therefore most of my feedback is to highlight the benefits of using such an approach, to someone less familiar with the approach.

Response: Thank you for your valuable feedback and suggestions. Our team acknowledges and agrees with the assessment. In the following text, we have addressed your comments in a point-by-point manner.

1. The results are interpreted in a valid way. However, it was unclear after reading the manuscript where they obtained data for incentive knowledge, or why they chose this particular variable, with appropriate literature showing its significance.

Response: We apologize for any confusion caused by the use of the term "knowledge of incentive policy." What we intended to convey was the "availability of incentive policy". The "availability of incentive policy" refers to the availability of monetary incentives such as rebates upon purchase and tax reduction after purchases [1], and non-monetary incentives such as high-occupancy vehicle (HOV) lane access for adopters [2]. We have made the necessary revision and changed all instances of "knowledge of incentive policy" to "availability of incentive policy."

We revised the manuscript accordingly.

- Section 1 Introduction: ~~To this end, intrinsic factors like individual demographics [3] [4] [5], knowledge of incentive policy [1] [6], charging access [7], as well as extrinsic factors like social media interactions [8] [9] have been identified as important to adoption decisions.~~ Specifically, innovation is relevant to vehicle-specific attributes such as purchase cost and maximum driving range [10], mass media or advertising regarding PEV [11], the built environment such as the availability of public charging infrastructure [12] and grid infrastructure [13], availability of monetary incentives such as rebate upon purchase and tax reduction after purchases [1], and non-monetary incentives such as high-occupancy vehicle (HOV) lane access for adopters [2].
- Section 2.2 Calibration of PEVs Diffusion Models: In detail, p relates to innovation-related factors like incentive policy availability and charging infrastructure access.
- Section 2.3 Parameters of PEVs Diffusion Models: ~~In detail, p relates to intrinsic factors like household income, knowledge about incentive policies, and access to charging stations.~~
- Section 2.3 Parameters of PEVs Diffusion Models: ~~For example, they have higher household incomes, they know more financial incentives policies, or they have more access to charging stations compared with the middle-income group and low-income group.~~
- Section 2.3 Parameters of PEVs Diffusion Models: It is widely recognized that vehicle-specific attributes [10], mass media [11], the built environment [12] [2], availability of incentives [1] are relevant to the willingness of consumers to accept PEVs.

2. The following statement could be discussed in further detail: "Altogether, this means that when the population is more sensitive to intrinsic factors, establishing new financial incentives for the whole potential market, increasing charging station availability, and reducing the upfront cost of adopting EVs will be more efficient. Otherwise, influencers recommend EVs to their friends or sending EVs to influencers will be more efficient." Specifically, there is literature which explores the different effects of charging infrastructure and financial incentives, with some finding that funding would be more effective if spent on charging infrastructure compared with incentives, and others finding that investments in home charging would be more effective than public charging. There is also a white paper by the ICCT titled "EXPANDING THE ELECTRIC VEHICLE MARKET IN U.S. CITIES" by Slowik et al. which explores the availability of incentives and infrastructure across U.S. cities and finds heterogeneity in cities with high PEV penetration and availability of incentives and charging.

Response: Thank you for sharing the ICCT white paper. We revised the statement and incorporated additional details into the discussion by including key conclusions from the ICCT white paper [2].

We revised the manuscript accordingly.

- Section 2.6 Promoting PEVs Adoption under Uncertainty: It is crucial to note that the scenario where imitation-related strategies have a more significant impact does not imply that policies and actions seeking a global effect do not matter. On the contrary, initiatives such as consumer incentives or the development of charging infrastructure are essential for facilitating adoption in low-income areas or regions with limited charging stations. However,

which are the best actions to increase the innovator rate is not clear. For example, Slowik *et al.* [2] assesses the PEV market in the 50 most populous metropolitan areas in the United States. They demonstrate that for areas with high PEV penetration, there is heterogeneity in the availability of policies, charging infrastructure, consumer incentives, and vehicle models. Altogether, Fig. 7 (B-D) shows that with different \hat{p}_{low} and \hat{q}_{low} effectively fitting the data, the year to reach critical adoption for different combinations of imitation-related strategy and innovation-related strategies varies as the contour of the year achieving critical adoption ranges from horizontally displayed and vertically displayed.

3. Authors could expand to clarify cases where lower-income households are more motivated intrinsically or extrinsically. It mentions above that in LA, influencer targeting is more effective for lower-income households, so would that mean they are more extrinsically motivated, and thus need less incentives and charging?

Response: We took Fig. 7 (B) and Fig. 7 (D) as examples to demonstrate cases where lower income has a relatively low fitted innovation coefficient or fitted imitation coefficient (We have changed the terms "intrinsic" and "extrinsic" to "innovation" and "imitation" based on your 7th comment). Besides, "influencer targeting is more effective for lower-income households", does not imply that they would not benefit from more incentives or charging stations. These improvements are also necessary to boost adoption.

We revised the manuscript accordingly.

– Section 2.6 Promoting PEVs Adoption under Uncertainty: Fig. 7 (B-D) compares innovation and imitation-related adoption boosting strategies, applied to the low-income group and for three pairs of \hat{p}_{low} and \hat{q}_{low} that fit current data. We represent the "innovation-related" strategy as an increase of \hat{p}_{low} in a fixed factor, which modifies the global propensity of innovation adoption within the low-income group. We represent the "imitation-based" strategy by setting the first n agents with the highest degree as adopters. Each figure shows years to critical adoption rates (16%) resulting from the innovation-related and the imitation-related strategies for each pair $(\hat{p}_{low}, \hat{q}_{low})$. Fig. 7 (B) illustrates that with a relatively low fitted \hat{p}_{low} and high \hat{q}_{low} , an imitation-related approach with more influencers can effectively boost adoption, whereas increasing the fixed factor (innovation-related strategy) does not have much effect. Conversely, Fig. 7 (D) depicts a scenario with a relatively high \hat{p}_{low} and low \hat{q}_{low} , where increasing fixed factors significantly enhances adoption, but adding more influencers has a negligible effect. Fig. 7 (C) presents a middle ground, where combining both strategies achieves the highest adoption increase. It is crucial to note that the scenario where imitation-related strategies have a more significant impact does not imply that policies and actions seeking a global effect do not matter. On the contrary, initiatives such as consumer incentives or the development of charging infrastructure are essential for facilitating adoption in low-income areas or regions with limited charging stations. However, which are the best actions to increase the innovator rate is not clear. For example, Slowik *et al.* [2] assesses the PEV market in the 50 most populous metropolitan areas in the United States. They demonstrate that for areas with high PEV penetration, there is heterogeneity in the availability of policies, charging infrastructure, consumer incentives, and vehicle models.

4. I was surprised that household income is the only demographic used to segment, considering the prominence of adopters living in/owning a detached home. I think this dimension is worth mentioning at the very least.

Response: We appreciate your thoughtful remarks. In our analysis, we only focused on income for two main reasons: (1) As we are in the initial phase of adoption, including too many variables increases the risk of overfitting. (2) Income is strongly correlated with other variables such as home type, ownership, and educational background. We added a note in the Supplementary Information to discuss other dimensions of demographics that could be used in segmentation (e.g., adopters living in/owning a detached home) and justify the market segmentation we used in this work.

We revised the manuscript accordingly.

- Section 4.1 Adopter Segmentation: We justify the choice of this segmentation approach in Supplementary Information (see Supplementary Information Note S7).
- Note S7: Justification of Market Segmentation: In this study, we divide the potential market based on income. Other related segmentation factors include educational background, home ownership, and housing type. In Fig. 1, we show that these three variables exhibit correlations. The figure illustrates the association of these three variables with income. Besides, with only three income groups, the fitted p and q already demonstrate significant variability due to limited data at the early adoption stage. Hence, despite the potential for more advanced applications to incorporate additional market segmentation with more socio-demographic parameters [3] or varying levels of the variables [14], we select income as the sole segmentation criterion.

Figure 1: Correlation plots of socio-demographic variables related to adoption. The income ratio represents the normalized income relative to the highest income across all tracts. (A,D) demonstrate correlation between the education ratio and income in Washington and California. The education ratio is defined as the proportion of the population aged 25 or older who have at least a Bachelor’s degree, relative to the total population aged 25 and older. (B,E) demonstrate correlation between ownership ratio and income in Washington and California. The ownership ratio refers to the percentage of owner-occupied units among all housing units. (C,F) demonstrate correlation between the housing ratio and income in Washington and California. The housing ratio indicates the percentage of detached single-family units among total housing units.

5. It may be helpful to highlight the benefits of the SocNet BM more in the introduction by incorporating more information that is presented in the first paragraph of section 2.2.

Response: Thank you for your suggestion. In response, we have emphasized the necessity of SocNet BM more in the introduction by incorporating additional details from the first paragraph to provide a clearer and more comprehensive overview.

We revised the manuscript accordingly.

- Section 1 Introduction: One limitation of the BM lies in its omission of spatial information and characterization of heterogeneity in social influence. The formulation of BM implies homogeneity and global interconnectedness, i.e., each agent’s probability of adoption is influenced uniformly by the adoption state of all other agents [15]. Another limitation of modeling adoption with the BM is that different socio-demographic groups are isolated in terms of social connections. To incorporate the role of geography and demographic characteristics in diffusion, we need models that consider both the spatial distribution of individuals and heterogeneity in the social network connections among different socio-demographic groups.
- Section 2.2 Calibration of PEVs Diffusion Models: ~~This formulation implies homogeneity and global interconnectedness, i.e., each agent’s probability of adoption is influenced uniformly by the adoption state of all other agents [15]. Another limitation of modeling adoption with the BMs is that different income groups are isolated in terms of social connections (see Methodology 4.2). To incorporate the role of geography and demographic characteristics in diffusion, we need models that consider heterogeneity in connections between locations with different socio-demographic groups.~~ This formulation implies homogeneity and global interconnectedness and fails to capture the heterogeneity in social connections among different socio-demographic groups from different locations.

6. Introduction could clarify what is meant by ”electric vehicle”. i.e., are authors combining BEV and PHEVs? surely there would be different power grid implications for BEVs vs. PHEVs, in addition to consumer profile and motivation.

Response: Thank you for bringing this issue to our attention. In this study, we focused on PEVs, which encompass both BEVs and PHEVs. We now recognize that the consumer profiles, motivations for adopting BEVs and PHEVs, and their implications for the power grid are distinct. We have added more discussion regarding this issue in the manuscript.

We revised the manuscript accordingly.

- Section 1 Introduction: This interplay is illustrated by the growing integration of plug-in electric vehicles (PEVs) –including both battery electric vehicles (BEVs) and plug-in hybrid vehicles (PHEVs)- into power systems.
- Section 3 Discussion: Last, this study uses data previous to 2023, but we note that the BEV stock in California was 3.8% by the end of 2023 [16], which crosses into early adopter territory. This allows us new opportunities to predict the diffusion in a more detailed way. The segmentation based on consumer characteristics alone is a valid method for forecasting PEV adoption [17]. However, consideration of the vehicle model is crucial to connect adoption with the power grid. Currently, due to limited data, we focus on PEVs without distinguishing

between BEVs and PHEVs. With more data, it will become feasible to predict the adoption of BEVs and PHEVs separately and to investigate the distinct implications for the power grid related to BEVs versus PHEVs, along with consumer profiles and motivations.

- We replaced all "EV" to "PEV".

7. "Intrinsic" and "extrinsic" could be more clearly defined, especially if it is being used in the introduction, and are the central components of the Bass model. Extrinsic also includes network effects, word of mouth, and marketing, not just social media interactions, as is currently stated in the intro, correct? Lee et al. (source 9) considers marketing to be an intrinsic factor. Perhaps more justification is needed on selection of factors, and perhaps being more explicit that these are the components of the BM.

Response: Thank you for your remarks. We now adhere to the original definitions in the Bass model [18] and have changed the terms "intrinsic" and "extrinsic" to "innovation-related" and "imitation-related." Innovation-related factors refer to the extent to which consumers are inclined to adopt PEVs independent of social influences, whereas imitation-related factors capture the word-of-mouth effects stemming from individuals who have already adopted or have experience with PEVs. We have supplemented the manuscript with further explanations to explicitly illustrate factors associated with components of the BM.

We agree that factors related to imitation extend beyond interactions on social media, including wider network effects and the dynamics of word-of-mouth. Besides, you are correct that mass marketing, as outlined in [3], is classified under "innovation-related" factors. However, influencer marketing, as discussed later in the paper, is distinct from mass marketing. Unlike mass marketing, influencer marketing accounts for the heterogeneity of social influences among adopters and non-adopters. We apologize for any confusion caused and have clarified the distinction between "mass marketing" and "influencer marketing" in the manuscript.

We revised the manuscript accordingly.

- Section 1 Introduction: These factors can be broadly categorized as innovation-related and imitation-related. Innovation-related factors determine the extent to which consumers are inclined to adopt PEV without the impact of social influences, whereas imitation-related factors pertain to word-of-mouth effects from individuals who have already adopted or have experience with the PEV. Specifically, innovation is relevant to vehicle-specific attributes such as purchase cost and maximum driving range [10], mass media or advertising regarding PEV [11], the built environment such as the availability of public charging infrastructure [12] and grid infrastructure [13], availability of monetary incentives such as rebate upon purchase and tax reduction after purchases [1], and non-monetary incentives such as high-occupancy vehicle (HOV) lane access for adopters [2]. Imitation, on the other hand, is shaped by social influences from friends [19], online interactions [20] as well as neighborhood and workplace interactions [21] [22].

- We replace "intrinsic" and "extrinsic" to "innovation-related" and "imitation-related".

8. Clarity on how knowledge of incentive policies was measured would be appreciated—financial incentives only? It would also be useful to justify the use of this variable,

as to how it would differ from perhaps availability of incentives? or take into account income caps?

Response: Thank you for your suggestions. By "knowledge of incentive policy," we meant "availability of incentive policy". We have now revised the manuscript to replace all instances of "knowledge of incentive policy" with "availability of incentive policy".

We revised the manuscript accordingly.

- Section 1 Introduction: ~~To this end, intrinsic factors like individual demographics [3] [4] [5], knowledge of incentive policy [1] [6], charging access [7], as well as extrinsic factors like social media interactions [8] [9] have been identified as important to adoption decisions.~~ Specifically, innovation is relevant to vehicle-specific attributes such as purchase cost and maximum driving range [10], mass media or advertising regarding PEV [11], the built environment such as the availability of public charging infrastructure [12] and grid infrastructure [13], availability of monetary incentives such as rebate upon purchase and tax reduction after purchases [1], and non-monetary incentives such as high-occupancy vehicle (HOV) lane access for adopters [2].
- Section 2.2 Calibration of PEVs Diffusion Models: In detail, p relates to innovation-related factors like incentive policy availability and charging infrastructure access.
- Section 2.3 Parameters of PEVs Diffusion Models: ~~In detail, p relates to intrinsic factors like household income, knowledge about incentive policies, and access to charging stations.~~
- Section 2.3 Parameters of PEVs Diffusion Models: ~~For example, they have higher household incomes, they know more financial incentives policies, or they have more access to charging stations compared with the middle-income group and low-income group.~~
- Section 2.3 Parameters of PEVs Diffusion Models: It is widely recognized that vehicle-specific attributes [10], mass media [11], the built environment [12] [2], availability of incentives [1] are relevant to the willingness of consumers to accept PEVs.

9. Context of what previous studies which use Bass model have found could improve the introduction. In addition to providing context of other method approaches, such as choice modeling, or regression models of disaggregate or aggregate data, for example, in addition to the Bayesian approach that is currently mentioned.

Response: Thank you for your comment. In response, we incorporated studies that have used the Bass model in the introduction. Additionally, we have now included key findings from methodologies using disaggregate and aggregate data to provide a more comprehensive overview of how previous research has approached the adoption problem.

We revised the manuscript accordingly.

- Section 1 Introduction: Previous studies establish a connection between aforementioned factors and the PEV diffusion projection utilizing aggregated and disaggregated data. The aggregated data is represented by the share of PEV vehicle sales at the country level [23], whereas the disaggregated data includes individual purchase records [24] [25] and surveys on travel behavior [26] or home charging access [5]. The spectrum of methodologies ranges from discrete choice models [27] to regression models [28]. For instance, Forsythe *et al.* [29] conduct a discrete choice experiment among new vehicle consumers in the United States.

This study indicates that projected advances in BEV range and price may allow them to equal or exceed the consumer valuation of gasoline vehicles by 2030. Wu *et al.* [30] employ Bayesian models to predict individual adoption probabilities and map the spatial distribution of charging demand of prospective adopters under varying adoption scenarios in the San Francisco Bay Area. They demonstrate that personalized shifting recommendations regarding charging schedules can reduce charging demand during power grid peak hours by 61% when achieving 90% PEV stock. Bayesian models can project the geographical distribution of future adopters given any adoption rate. However, it cannot output the time to reach such adoption. To provide temporal projections, the Bass model (BM) is widely used. The BM analyzes the market as a whole using two parameters (related to imitation and innovation factors, respectively). In 2009, Becker *et al.* [31] incorporate assumptions related to oil price fluctuations and incentives with BM and project that 65% of vehicle sales by 2030 will consist of PEVs. Jenn *et al.* [32] utilize sales data from 39 countries and explore PEV market expansion through BM. They find that it may be challenging to achieve targets such as 100 million electric vehicles worldwide by 2030. Lee *et al.* [3] use BM to predict the adoption dynamics of heterogeneous groups of PEV buyers. Their findings indicate that although high-income families currently constitute 49% of the PEV market, they form only 3.6% of households in California, suggesting this group may not remain the dominant PEV adopters. To sustain market growth, policymakers should account for the infrastructure and incentive requirements of consumers from mid/high- and middle-income groups. Despite the ability to provide temporal projections, BM does not provide the spatial distribution of future adopters. One limitation of the BM lies in that the formulation of BM implies homogeneity and global interconnectedness, i.e., each agent’s probability of adoption is influenced uniformly by the adoption state of all other agents [15]. This leads to the omission of spatial information and characterization of heterogeneity in social influence. Another limitation of modeling adoption with the BM is that the model represents each demographic group in isolation and thus disregards any interaction the groups may have. To incorporate the role of geography and demographic characteristics in diffusion, we need models that consider both the spatial distribution of individuals and heterogeneity in the social network connections among different socio-demographic groups.

10. Introduction could overall be improved by providing context of various mandates of EV sales, which further motivates the need to model the effects on power grid, in addition to being more specific about the socio-demographic profile of early PEV adopters.

Response: Thank you for your feedback. In response, we have included information on the mandates for EV sales worldwide. Additionally, we have specified the socio-demographic profiles of early PEV adopters to offer further insight into the characteristics of individuals who are likely to adopt PEVs in the early stages.

We revised the manuscript accordingly.

- **Section 1 Introduction:** PEVs offer significant environmental advantages by reducing traffic-related air pollutants and cutting down greenhouse gas emissions [33] [34]. In addition, they provide economic benefits such as improved fuel efficiency and enhanced energy security [35]. Recognizing these advantages, the United States has enacted policies to encourage the uptake of PEVs. In 2022, the California Air Resources Board (CARB) approved the Advanced Clean Cars II regulation to ensure that by 2035, all new cars and light trucks sold

in California will be zero-emission vehicles (ZEVs) [36]. That same year, Washington state announced it would follow California and prohibit the sale of new gas-powered vehicles by 2035 [37]. Meanwhile, worldwide efforts to accelerate the adoption of PEVs are exemplified by the China mandate for new energy vehicles [38], the United Kingdom’s plan to transition to ZEVs [39], and the European Union’s CO2 emissions regulations [40]. These initiatives are expected to substantially boost adoption in the near future. Aligned with this trend, modeling the geographical distribution of future PEV adopters becomes a pivotal task. This is because human decisions on when and where to adopt PEVs are linked directly to the regional electricity demands [41], the consequent pressure on the power grid [42], and the power system infrastructure planning [43].

- **Section 1 Introduction:** The diffusion of PEVs exhibits significant difference across different demographic groups at the early stage. Evidence from the global PEV market indicates that early PEV adopters tend to have higher incomes [44], advanced education [45], and reside in non-apartment houses with home charging access [12].

11. There seems to be a disconnect between the introduction emphasizing implications for the power grid, and the discussion not mentioning this at all.

Response: Thank you for your remark. We have added more detailed implications for the power grid in the discussion section.

We revised the manuscript accordingly.

- **Section 3 Discussion:** Our findings also provide implications for power system planning. Primarily, PEV adoption is uneven across regions. This study provides benchmark to predict geographical distribution of PEV adopters. In areas with high PEV adoption, the stress on transmission and distribution networks intensifies. If the charging of these adopters remains uncoordinated, it may cause transformer overloading [46], overload transmission lines [47], early replacement of equipment [48]. This requires strategic control and infrastructure build-out to avoid bottlenecks and maintain reliable service [42]. Additionally, given demographic disparities in EV adoption, equitable planning for charging infrastructure is crucial. The results show that middle- and low-income groups are less inclined to adopt PEVs due to innovation-related factors compared to the high-income group. To encourage PEV adoption among these communities, grid operators may need to offer lower electricity rates or enhance grid capacity to support new charging stations nearby. Moreover, the uncertainty in adoption forecasts demands integrating distributed energy resources, such as energy storage [49], renewables [50], vehicle-to-grid solutions [51] to bolster grid resilience.

12. Reference 7 is about fleet decisions, not sure if it’s appropriate since the scope of study is on consumer decisions? Further, on the same line, I believe ”choices of early adopters” could be improved to be more specific.

Response: Thank you for your careful reading and suggestion. We have taken out reference 7 and updated the sentence about the ”choice of early adopters” to clarify that ”choice of early adopters” refers to ”whether to adopt PEV”.

We revised the manuscript accordingly.

- Section 1 Introduction: ~~Researchers start by understanding the choices of the early adopters [12] [52].~~ The diffusion of PEVs exhibits significant differences across different demographic groups at the early stage.

13. References 9,10, & 11 could be specific about what kinds of demographics are related to BEV adoption. For example, source 11 is about home charge access; additionally, it may be appropriate to differentiate between home and public charging access (reference 14).

Response: Thank you for your suggestion. We have revised the sentence to specify which demographics are related to PEV adoption. We also differentiate between home and public charging access in the manuscript.

We revised the manuscript accordingly.

- Section 1 Introduction: ~~To this end, intrinsic factors like individual demographics [3] [4] [5], knowledge of incentive policy [1] [6], charging access [7], as well as extrinsic factors like social media interactions [8] [9] have been identified as important to adoption decisions.~~ Evidence from the global PEV market indicates that early PEV adopters tend to have higher incomes [44], advanced education [45], and reside in non-apartment houses with home charging access [12].
- Section 1 Introduction: Specifically, innovation is relevant to vehicle-specific attributes such as purchase cost and maximum driving range [10], mass media or advertising regarding PEV [11], the built environment such as the availability of public charging infrastructure [12] and grid infrastructure [13], availability of monetary incentives such as rebate upon purchase and tax reduction after purchases [1], and non-monetary incentives such as high-occupancy vehicle (HOV) lane access for adopters [2].

14. I do not believe that references 12 & 13 are appropriate citations for "knowledge of incentive policy". Reference 12 measures the effect of financial incentives for PEV adopters, and reference 13 is about the effect of various policy scenarios on business strategy; neither are within the scope of measuring consumer knowledge of incentives. To that end, I'm not sure reference 13 is appropriate for a paper focusing on consumer decisions.

Response: Thank you for your suggestion. The term "knowledge of incentive policy" has been changed to "availability of incentive policy" and references 12 and 13 have been removed. Additionally, we have included new references to clarify the "availability of incentive policy".

We revised the manuscript accordingly.

- Section 1 Introduction: ~~To this end, intrinsic factors like individual demographics [3] [4] [5], knowledge of incentive policy [1] [6], charging access [7], as well as extrinsic factors like social media interactions [8] [9] have been identified as important to adoption decisions.~~ Specifically, innovation is relevant to vehicle-specific attributes such as purchase cost and maximum driving range [10], mass media or advertising regarding PEV [11], the built environment such as the availability of public charging infrastructure [12] and grid infrastructure [13], availability of monetary incentives such as rebate upon purchase and tax reduction after purchases [1], and non-monetary incentives such as high-occupancy vehicle (HOV) lane access for adopters [2].

15. Regarding extrinsic motivation (reference 15 and 16), would not in-person interactions also be considered extrinsic? Perhaps more references are needed for word-of-mouth, or workplace network effects.

Response: Thank you for your suggestion. We agree that "imitation-related" factors would also encompass in-person interactions, word-of-mouth, and workplace network effects. We have expanded the definition of "innovation-related" factors.

We revised the manuscript accordingly.

- Section 1 Introduction: ~~To this end, intrinsic factors like individual demographics [3] [4] [5], knowledge of incentive policy [1] [6], charging access [7], as well as extrinsic factors like social media interactions [8] [9] have been identified as important to adoption decisions.~~ Imitation, on the other hand, is shaped by social influences from friends [19], online interactions [20] as well as neighborhood and workplace interactions [21] [22].

16. I do not believe references 32 and 33 are appropriate for the statement that "adoption choices are related to household income" 32 is regarding the business case and supply chain for PEVs, the other is mostly about being "eco-friendly", with income not being mentioned in the abstract.

Response: Thank you for your suggestion. We have removed references 32 and 33, and added references [44], [3], and [45] to support the claim that adoption choices are related to household income.

We revised the manuscript accordingly.

- Section 4.1 Market Segmentation: Previous research about understanding the driving forces behind PEV adopters' decisions highlights that adoption choices are related to household incomes ~~[53] [54] [44] [3] [45]~~. Therefore, we divide the potential market into three socio-demographic groups: low-income group (income below 33% of the reference population of the state), middle-income group (income between 33% and 66% of the reference population of the state), and high-income group (income above 66% of the reference population of the state).

17. References could overall be improved by more robustly choosing appropriate literature for each component of the BMs, and incorporating the literature into the discussion, as there are currently no references in the discussion.

Response: Thank you for your suggestion. In response, we have expanded the introduction to include literature on each component of the BMs. These factors can be broadly categorized as innovation-related and imitation-related. Additionally, we have added more implications for EV adoption from the power grid perspective and discussed possible future directions based on the current PEV adoption trend. This should help to better contextualize the findings in terms of their broader implications.

We revised the manuscript accordingly.

- Section 1 Introduction: The diffusion of PEVs exhibits significant differences across different demographic groups at the early stage. Evidence from the global PEV market indicates

that early PEV adopters tend to have higher incomes [44], advanced education [45], and reside in non-apartment houses with home charging access [12]. In this regard, forecasting PEV adoption involves addressing the complex challenge of modeling the impact of various factors on adoption across diverse demographic segments. These factors can be broadly categorized as innovation-related and imitation-related. Innovation-related factors determine the extent to which consumers are inclined to adopt PEV without the impact of social influences, whereas imitation-related factors pertain to word-of-mouth effects from individuals who have already adopted or have experience with the PEV. Specifically, innovation is relevant to vehicle-specific attributes such as purchase cost and maximum driving range [10], mass media or advertising regarding PEV [11], the built environment such as the availability of public charging infrastructure [12] and grid infrastructure [13], availability of monetary incentives such as rebate upon purchase and tax reduction after purchases [1], and non-monetary incentives such as high-occupancy vehicle (HOV) lane access for adopters [2]. Imitation, on the other hand, is shaped by social influences from friends [19], online interactions [20] as well as neighborhood and workplace interactions [21] [22].

- **Section 3 Discussion:** This study also provides implications for power system planning. Primarily, PEV adoption is uneven across regions. This framework provides a benchmark to predict the geographical distribution of PEV adopters. In areas with high PEV adoption, the stress on transmission and distribution networks intensifies. If the charging of these adopters remains uncoordinated, it may cause transformer overloading [46], overload transmission lines [47], early replacement of equipment [48]. This requires strategic control and infrastructure build-out to avoid bottlenecks and maintain reliable service [42]. Additionally, given demographic disparities in PEV adoption, equitable planning for charging infrastructure is crucial. The results show that middle- and low-income groups are less inclined to adopt PEVs due to innovation-related factors compared to the high-income group. To encourage PEV adoption among these communities, grid operators may need to offer lower electricity rates or enhance grid capacity to support new charging stations nearby. Moreover, the uncertainty in adoption forecasts demands integrating distributed energy resources, such as energy storage [49], renewables [50], vehicle-to-grid solutions [51] to bolster grid resilience.

~~This work could be extended in many directions. One way could be to develop methods to infer social network structure influencing the adoption by collecting that information from the new adopters. An interesting addition to the model would be to include the effects of charging infrastructure locations and tax deduction policies.~~ This research may be expanded in various ways. For instance, integrating technology, policy, and built-environment evolution into diffusion predictions could be one area of focus. Alternatively, planning for power and transportation infrastructure in response to varying forecasts of PEVs adoption represents another avenue. Lastly, this study uses data prior to 2023, but we note that the BEV stock in California was 3.8% by the end of 2023 [16], which crosses from the innovator phase into the early adopter phase. Larger numbers of adopters will provide new opportunities to predict the diffusion in a more detailed way. For example, while segmentation based on consumer characteristics alone is a valid method for forecasting PEV adoption [17], the consideration of the vehicle models is crucial to connect adoption with the power grid. Currently, due to limited data, we focus on PEVs without distinguishing between BEVs and PHEVs. With more data, it will become feasible to predict the adoption of BEVs and PHEVs separately and to investigate the distinct implications for the power grid related to BEVs versus PHEVs, along with consumer profiles and their motivation.

Comments from Reviewer 2

This study proposes an innovative approach that integrates spatial information and social network into a technology diffusion model to predict long-term EV adoption. The findings are significantly different from existing studies, showing a much more conservative adoption rate. This finding emphasizes the importance of including spatial and social network structures in EV technology diffusion and adoption prediction. The paper is well-written and well-structured. However, several comments need to be addressed before publishing.

Response: Thank you for your remarks and suggestions. We have carefully revised the manuscript to address these concerns, and the responses are provided point-by-point below.

1. One significant limitation of the paper is the lack of consideration of the possibility that EV users might switch back to gasoline vehicles. This study assumes that once consumers adopt EVs, they will not switch back, overlooking the behavioral uncertainty in the market. Many factors such as price changes in fuel/electricity, charging infrastructure availability, and shifts in consumer preferences could lead to a technology reverse migration. Report shows that about 46% of current EV users in U.S. are likely to switch back to ICE vehicles https://executivedigest.sapo.pt/wp-content/uploads/2024/06/Mobility-Consumer-Pulse-2024_Overview.pdf. Such impact has been included in the data (e.g., number of EV titles) but the model ignored. The omission of this reverse migration could lead to an overestimation of EV adoption. This possibility should be incorporated into the modeling framework to provide a more comprehensive forecast of EV adoption. For example, in the simulation process, each agent that adopted EV should be assigned a probability of abandoning EV after several years of usage.

Response: Thank you for sharing the Mobility Consumer Pulse 2024 Report, and we agree with you that it is important to consider the possibility of people switching back to gasoline cars. In the main manuscript, we focus on presenting the most optimistic scenarios for future PEV adoption. In the supplementary information, we provide additional experiments to illustrate the probability of abandoning PEVs after several years of use. This allows us to capture a more comprehensive range of potential outcomes and better address the complexities of long-term adoption trends.

We revised the manuscript accordingly.

- Section 2.4 Divergence of PEVs Adoption Forecasts: We forecast EV adoption rates from 2023 to 2050 with the calibrated models. In this section, we present the most optimistic adoption forecast by assuming that PEV users will not revert to internal combustion engine vehicles (ICEVs). The forecast considering the potential switching back to ICEVs by PEV users is deferred to Supplementary Information (see Supplementary Note S4).
- Note S4: Note S4: Forecast Considering “Switching Back to Internal Combustion Engines Vehicles (ICEVs)”: Various elements—including fluctuations in fuel/electricity prices, the availability of charging stations, and changes in consumer tastes—may cause a reversal towards older technology. According to [55], approximately 46% of current PEV users in the United States may consider reverting to ICE vehicles. Our model allows for the simulation of “Switching Back to ICEVs” by assigning a likelihood that users will abandon PEVs af-

ter several years of use. We examine different scenarios by adjusting two parameters: *Year* indicates the number of years before users start to consider transitioning back to ICEVs, while *Prob* expresses the probability of this switch. For example, “*Year* = 5 and *Prob* = 0.5” indicates that after 5 years of adoption, the adopter will have a probability of 0.5 of switching back to ICEVs every year thereafter. Our settings include *Year* = 5, 10, 15 and *Prob* = 0.1, 0.3, 0.5. Setting “No Return” represents the baseline where customers do not switch back to ICEVs. Fig. 2 (A-B) demonstrates how a smaller *Year* and a larger *Prob* slows down the PEV diffusion in Los Angeles.

Figure 2: PEV diffusion forecast accounting for “Switching Back to ICEVs”. (A) demonstrates adoption forecast based on varying timeframes (in years) before users begin considering a transition back to ICEVs (*Year* set as 5). (B) demonstrates adoption forecast based on different probabilities of switching back to ICEVs (*Prob* set as 0.5).

2. For the method part, the reviewer has one concern in Section 4.3.1, where the authors arbitrarily choose $n=50,000$ nodes to construct the social network in two different cities. However, it lacks a clear rationale for this selection. It also raises concerns about the robustness of the results. First, the paper does not provide any rationale for why this specific number of nodes was chosen, nor does it discuss how this choice might affect the model’s predictions. The number of nodes in a social network model is a critical parameter that may significantly affect the simulation process. Second, different cities have varying population sizes, social structures, and urban layouts. All of them require different network sizes to capture the heterogeneity. Assigning 50,000 nodes for different regions of interest oversimplifies the social structures. For example, a city with a larger population or more complex human interactions might require a denser or larger network to represent the social structure. When stakeholders from other regions/cities want to apply the model from this study, it is not clear how to choose the appropriate node count for a better prediction. Third, without a sensitivity analysis, it is unclear whether the model’s results are reliable or whether they could change significantly with a different number of nodes. This uncertainty weakens the model’s prediction performance and limits its applicability to real-world scenarios.

Response: We appreciate your valuable feedback, and we agree that justifying the number of nodes is critical. In the revised manuscript, we now determine the number of nodes for each county by fixing the expansion factor, which is defined as the empirical number of vehicles from the census / number of nodes. The expansion factor ranges from 1 to the total number of vehicles, with the expansion factor of one ideally representing the most accurate reflection

of reality. To balance computational time and the efficiency of the framework, we selected the smallest expansion factor that ensures the number of nodes does not compromise the robustness of the results. After testing different factors, we selected an expansion factor of 5 (Before the expansion factor is 100 for Los Angeles).

We revised the manuscript accordingly.

- Section 4.3.1 Social Network Construction: For the region of interest, ~~we have $n = 50,000$ nodes in the network, where each node represents a potential adopter.~~ we sample nodes from data with expansion factor = 5 by keeping spatial distribution (an expansion factor is defined as the empirical number of vehicles from census / number of nodes). The sensitivity analysis for various expansion factors is available in the Supplementary Information (see Supplementary Information Note S8).
- Note S8 Sensitivity Analysis on Network Size: This sensitivity analysis illustrates the impact of varying network sizes on the social network structure, adoption rate, and Moran’s I for the adoption rate. We configure various network sizes by selecting expansion factors of 5, 10, 15, 25, and 100 in Los Angeles. As presented in Table 1-3, an increase in the expansion factor results in reduced network size, characterized by fewer nodes and edges. Our analysis also demonstrates how average degree, average geographical distance between two nodes, clustering coefficient, and assortativity vary across different expansion factors.

Expansion Factor	# Nodes	# Edges	Average Degree	Average Distance	Clustering Coefficient	Assortativity
5	1,189,150	3,567,441	6	21.67	0.0001	0.07
10	594,480	1,783,431	6	21.68	0.0002	0.07
15	396,364	1,189,083	6	21.66	0.0002	0.07
25	237,827	713,472	6	21.70	0.0004	0.07
100	59,453	178,350	6	21.76	0.0012	0.07

Table 1: Expansion factor and corresponding social network structure summary ($\gamma = 1$)

Expansion Factor	# Nodes	# Edges	Average Degree	Average Distance	Clustering Coefficient	Assortativity
5	1,189,150	3,567,441	6	1.95	0.01	0.65
10	594,480	1,783,431	6	1.96	0.02	0.65
15	396,364	1,189,083	6	1.97	0.02	0.65
25	237,827	713,472	6	1.99	0.03	0.64
100	59,453	178,350	6	2.14	0.08	0.62

Table 2: Expansion factor and corresponding social network structure summary ($\gamma = 3$)

We further illustrate the impact of varying expansion factors on adoption rates and Moran’s I. Figure 3 (A-C) shows that smaller expansion factors result in higher Moran’s I, whereas Figure 3 (D-F) indicates that the expansion factor has minimal effect on adoption rates. Consequently, we decide to use the smallest feasible expansion factor, which is 5.

Expansion Factor	# Nodes	# Edges	Average Degree	Average Distance	Clustering Coefficient	Assortativity
5	1,189,150	3,567,441	6	0.71	0.04	0.96
10	594,480	1,783,431	6	0.71	0.07	0.96
15	396,364	1,189,083	6	0.72	0.10	0.95
25	237,827	713,472	6	0.72	0.14	0.95
100	59,453	178,350	6	0.78	0.33	0.92

Table 3: Expansion factor and corresponding social network structure summary ($\gamma = 10$)

Figure 3: Network expansion factors' influence on diffusion patterns. The dots present the median results of multiple repetitions of simulation using a predefined p and q set. (A-C) demonstrate the correlation of adoption rates and corresponding Moran's I across various expansion factors (Expansion Factor = 5, 10, 25, 100) and different network structures ($\gamma = 1, 3, 10$). (D-F) demonstrate adoption rates evolution across various expansion factors (Expansion Factor = 5, 10, 25, 100) and different network structures ($\gamma = 1, 3, 10$).

3. It is unclear whether/how real world location data is used in the social network construction. Because SocNet BM is the major contribution of this work, it could benefit from having more technical details (e.g., in the Supplementary Information) to improve method clarity.

Response: Thank you for your suggestion. We have included a note in the Supplementary Information to clarify how real-world location data is utilized in constructing the social network.

We revised the manuscript accordingly.

- Section 4.3.1 Social Network Construction: These nodes are in a two-dimensional space described by the longitude and latitude of potential adopters’ home locations (the geographical resolution is at census tract level). Then we can calculate the distance between any adopter i and adopter j as d_{ij} . Once the nodes are distributed in this space, ~~we have to construct the links and we use the following algorithm: (1) Selected at random a subset of n_0 initial active nodes. (2) Take an inactive node i at random and connect it with an active node j with probability (up to a normalization factor) $prob_{i \rightarrow j} \propto \frac{K(k_j)}{D(d_{ij})}$, where k_j is the connectivity of node j , d_{ij} is the distance in kilometers between nodes i and j , and $K(\cdot)$ and $D(\cdot)$ are given functions. Finally (3), make the node i active and go back to (2) until all nodes are active. For each node, we repeat $k_r = 3$ times the steps (2)-(3) so that the average connectivity will be $\langle k \rangle = 2k_r$. To include preferential attachment and distance selection, we use $K(k) = k + 1$ and $D(d) = d^\gamma$, where $\gamma = 1, 3, 10$.~~ we construct links using the following algorithm: (1) Select at random a subset of n_0 initial active nodes. (2) Take an inactive node i at random and connect it with an active node j with probability:

$$prob_{i \rightarrow j} \propto \frac{K(k_j)}{D(d_{ij})}, \quad (1)$$

where k_j is the connectivity of node j , d_{ij} is the distance in kilometers between nodes i and j , $K(\cdot)$ and $D(\cdot)$ are given functions, and $prob_{i \rightarrow j}$ is then normalized up to a factor $\sum_{j \in \text{active nodes}} prob_{i \rightarrow j}$. For each inactive node, we sample $k_r = 3$ edges so that the average connectivity will be $\langle k \rangle = 2k_r$, and make the node i active. Finally (3), repeat (2) until all nodes are active. To generate preferential attachment and empirical spatial features, we use $K(k) = k + 1$ and $D(d) = d^\gamma$, where $\gamma = 1, 3, 10$ (see Supplementary Information Note S9 for the rationale behind the selection of γ values). We illustrate the algorithm with an example in the Supplementary Information (see Supplementary Information Note S10).

- Note S10 Social Network Construction Process: Fig. 4 depicts the procedure for constructing a social network. Subfigures (A-C) illustrate the assignment of geographical information to nodes based on census data [56], while (D-F) illustrate the process of forming edges among nodes.

Figure 4: Social network construction process. (A) Number of vehicles in each census tract (obtained from the census tract data). (B) Each node represents one node (when the expansion factor is 1). The latitude and longitude for each node are set as centroid coordinates of its respective census tract. (C) Based on the latitude and longitude for each node, the distance d_{ij} between any two nodes, i and j , can be computed. For nodes residing within the same census tract, the distance is set as half the square root of the census tract's area. (D) Nodes 1, 2, and 6 are selected as the active nodes set, leaving nodes 3, 4, and 5 as inactive. (E) Inactive node 4 is randomly chosen, and $prob_{4 \rightarrow j}$ is calculated and normalized where $j \in \{1, 2, 6\}$. For example, for node 4 and node 1, $k_4 = 0$ and d_{41} is the distance between the centroid of census tract 4 and census tract 1. Based on $prob_{4 \rightarrow j}$, $m = 3$ edges are created linking node 4 to nodes in the active nodes set. Node 4 is then marked as active. This step is repeated for the remaining inactive nodes until all are active. (F) This process is executed for nodes 3 and 5 to finalize the social network.

4. Figure 5 (C,D,E) indicates that the urban layout could be a very important factor. It directly affects both the spatial and social structures of a city/county. Different regions/cities have varying urban designs, including different grid sizes and population densities. For example, densely packed urban areas could have quicker adoption because of strong geographical and social connections, while sprawling suburban areas may show slower diffusion due to greater distances between adopters. The urban layout is highly correlated with spatial and social structure, which further can affect EV adoption. The reviewer expects to see more analysis and discussion about the correlation between urban layout and adoption diffusion patterns. Understanding this correlation can provide more insights into future EV adoption dynamics

Response: Thank you for your valuable suggestion. We agree that the urban layout directly influences the spatial and social structures of an area, which in turn can affect PEV adoption. In the supplementary information, we have added a note to highlight how urban layout impacts the diffusion process. This includes an analysis of the correlation between urban spatial/social structures and adoption diffusion patterns.

We revised the manuscript accordingly.

- Section 2.5 Effects of Network Structure on Spatial Diffusion: Specifically, in Los Angeles, $\gamma = 10$ appears to more accurately characterize the social network associated with PEV adoption (see Supplementary Information Note S6 for results in other counties). Overall, the SocNet BM with spatial social network allows more flexibility to capture the spatial autocorrelation of the empirical diffusion.
- Note S6 Urban Layout and Diffusion Process: The urban layout of the area directly affects the spatial and social structures of areas, which can further affect PEV adoption. This section provides an analysis of the correlation between urban spatial/social structures and adoption diffusion patterns. Regarding spatial structure, we consider the geographical autocorrelation of population density, the geographical autocorrelation of income, and the overall population density at the county level. Concerning the social structure of areas, we take into account various γ values that determine the social network structure.

In terms of spatial structure, Fig. 5 demonstrates that counties with greater population and higher geographical autocorrelation in terms of income and population density typically exhibit higher adoption rates in present and forthcoming adoption trends.

In terms of social structure, Fig. 6 (A-C) display the Moran's I statistics for State/County BM and SocNet BM across three counties in Washington, while Fig. 6 (D-F) show the same in California. In California and Washington's most populous counties, specifically King and Los Angeles, we can identify a social network structure parameter (γ) that closely replicates the observed Moran's I statistics. Nonetheless, counties such as Santa Clara and Pierce have no γ value that can accurately replicate the empirical Moran's I statistics.

Figure 5: Spatial structure and diffusion process. Each circle represents one county in Washington and California. The solid line denotes the linear regression results of circles. (A-C) plot the county-level adoption rate and corresponding population density Moran's I, Income Moran's I, and overall population density in 2022. (D-F) plot county-level adoption rate and corresponding population density Moran's I, Income Moran's I, and overall population density in 2050.

Figure 6: Moran's I statistics for State/County BM and SocNet BM in 6 counties in Washington and California. For SocNet BM, we use the network with three γ s, i.e., 1,3,10. The fitting process is repeated for each γ and the median of predictions is reported. Solid lines with circle markers show the median (50th percentile) Moran's I statistics from SocNet BM. Dashed lines with cross markers and star markers show the Moran's I statistics from State/County BM; Squares show the Moran's I statistics calculated from empirical data on tract level.

5. When evaluating the adoption rate error for 2022, which years of data were used to fit the model? Also, it is unclear whether the “adoption rate error” values are absolute values in terms of adoption percentage or percentage values relative to the ground truth data. Please clarify. It would also be helpful to show the overall performance of each model for the entire state, considering that some models may overperform others in some counties but underperform in others.

Response: We use data from 2010 to 2022 to fit the model and have demonstrated the state-level fitted error from 2010 to 2022, as well as the county-level and tract-level error for 2022 in Figure 2. The adoption rate error values are calculated as the absolute difference between the simulated adoption rate and the ground truth adoption rate in 2022. Additionally, we have included the overall performance of each model for the entire state in the manuscript.

We revised the manuscript accordingly.

- Section 2.2 Calibration of PEVs Diffusion Models: On the tract level, Fig. 2 (B, D) show the distribution of adoption rate error for three models. The adoption rate error values are the absolute difference between the simulated adoption rate and ground truth adoption rate in 2022.
- Section 2.2 Calibration of PEVs Diffusion Models: All models fit similarly well with the empirical data, i.e., State/County BM has a mean absolute error of 0.03% and SocNet BM of 0.08% in Washington. In California, the errors are 0.08% and 0.11%, respectively.

6. The code link is not valid. Please double-check.

Response: Thank you for pointing out this issue. We apologize for any inconvenience caused. The code and data can be accessed via the following link: github.com/humnetlab/Planning-the-Electric-Vehicle-Transition-by-Integrating-Spatial-Information-and-Social-Networks.

We revised the manuscript accordingly.

- Code and Data Availability: The code and data used for the analysis presented in this paper are available at GitHub: github.com/humnetlab/Planning-the-Electric-Vehicle-Transition-by-Integrating-Spatial-Information-and-Social-Networks. Please contact M.C.G (martag@berkeley.edu) and J. W. (jmwu@berkeley.edu) for any questions.

7. Page 4. Most of the content in the paragraph “Fig. 2 (A, C) show ...” has been included in the figure caption. The figure caption is informative enough to understand the figure independently. In addition, it is not clear the meaning of the filled area between two curves in Fig. 2 (A, C). More explanation should be included.

Response: Thank you for your advice. We have removed the redundant information from the text. The filled area between the two curves in Fig. 2 (A,C) represents the 5th to 95th percentile of the state-level adoption rate.

We revised the manuscript accordingly.

- Section 2.2 Calibration of PEVs Diffusion Models: Fig. 2 (A,C) show the state-level aggregated adoption rate ~~from State/County BM, SocNet BM, and empirical data~~ in Washington and California between 2010 and 2022. ~~The different colors represent three income levels from darker to lighter, going from higher to lower income groups respectively.~~

8. In Figure 2 (E, F), the data points overlap with the box, making it hard to compare the model fitness of different counties. Separating the box and data points into two columns for each county can improve the readability of this figure.

Response: Thank you for your suggestion. To improve the readability, we have separated the box and data points into two columns for each county.

We revised the manuscript accordingly.

- Section 2.2 Calibration of PEVs Diffusion Models: We revised Fig. 7 as follows.

Figure 7: Comparison of State/County BM and SocNet BM's performance. (A, B) State-level fitted adoption curves of State BM (cross), County BM (star), SocNet BM (circle), and data (square) for low (light blue/yellow), middle (blue/orange), and high (dark blue/red) income groups in Washington and California from 2010 to 2022. Solid lines with markers show the median (50th percentile) adoption rate of income groups in states; the shaded bands show the range from the 5th to 95th percentile, highlighting the uncertainty. (C, D) Distribution of tract level adoption rate error of State BM (cross), County BM (star), and SocNet BM (circle) in Washington (blue) and California (orange) in 2022. (E, F, D, E) County-level adoption rates of State/County BM and SocNet BM in the most populated 20 15 counties in Washington (blue) and California (orange) in 2022. The box shows the quartiles of the county-level adoption rates of SocNet BM while the whiskers extend to show the rest of the distribution. Cross markers show the adoption rate of State BM, star markers show the adoption rate of County BM, and square markers show the empirical data.

9. The explanation of parameters p and q in the first paragraph of section 2.3 should be integrated with the similar content in section 2.2.

Response: Thank you for your suggestion. We have integrated the explanation of parameters p and q from section 2.3 into section 2.2 to reduce redundancy and streamline the presentation.

We revised the manuscript accordingly.

- Section 2.2 Calibration of PEVs Diffusion Models: We use the BMs at the state level and the county level as benchmarks, referred here as State/County BM (see Methodology 4.2). ~~These models fit the adoption curves in time at the referred geographies. Yet they are not built on a social network with spatial information. Such information is essential for modeling the geographical distribution of future adopters.~~ In BMs, the probability of every agent to adopt, given that it has not adopted so far, depends linearly on an ~~intrinsic~~ innovation-related influence p and an ~~extrinsic~~ imitation-related influence q that depends on the fraction of prior adopters [18]. ~~This formulation implies homogeneity and global interconnectedness, i.e., each agent's probability of adoption is influenced uniformly by the adoption state of all other agents [15]. Another limitation of modeling adoption with the BMs is that different income groups are isolated in terms of social connections (see Methodology 4.2). To incorporate the role of geography and demographic characteristics in diffusion, we need models that consider heterogeneity in connections between locations with different socio-demographic groups. This formulation implies homogeneity and global interconnectedness and fails to capture the heterogeneity in social connections among different socio-demographic groups from different locations.~~
- Section 2.3 Parameters of PEVs Diffusion Models: ~~The BM represents the adopter amount at a certain time by coefficients p and q [18]. p can be seen as the rate of innovators' adoption, and q is the rate of imitation of adoption. Innovators' adoption is not affected by others, while imitators depend upon the social interaction with these innovators. Initially, only p matters; subsequently, once the innovation adoption starts, q takes over the adoption. These two parameters are closely connected to intrinsic and extrinsic factors. In detail, p relates to intrinsic factors like household income, knowledge about incentive policies, and access to charging stations. q relates to extrinsic factors like word-of-mouth, social networking, and marketing.~~

10. Typo: “Our comparison of projections for future EV adopters reveals that by introducing social network, the number of adopters by 20250 is 25% of benchmark”, which should be “2050”.

Response: Thank you for your careful reading. We have revised the sentence.

We revised the manuscript accordingly.

- Section 3 Discussion: Our comparison of projections for future EV adopters reveals that by introducing social network, the number of adopters by ~~20250~~ 2050 is 25% of benchmark projections without network structure.

11. Figures in the Supplementary Information could benefit from better caption and legend formatting.

Response: Thank you for your suggestion. We have revised the caption and formatting of all figures in the Supplementary Information for improved clarity and consistency.

We revised the manuscript accordingly. Two examples of caption modifications are as follows.

- Note S1 Fitting Error on County Level: Figure 1: ~~County-level fitted adoption rate in 2022 for three models.~~ Figure 1: County-level adoption rates of data, State/County BM and SocNet BM in 2022. The box shows the quartiles of the county-level adoption rates of SocNet BM while the whiskers extend to show the rest of the distribution. Cross markers show the adoption rate of State BM, star markers show the adoption rate of County BM, and square markers show the empirical data. (A-B) demonstrate results in Washington (blue) and (C-D) demonstrate results in California (orange).
- Note S2 Fitting Error on Tract Level: Figure 2: ~~Tract-level fitted adoption rate in 2022 for three models.~~ Tract-level adoption rate comparison between empirical data and simulations of State BM (orange), County BM (dark blue), and SocNet BM (light blue) in 2022. Each point presents the fitted tract-level adoption rate and the corresponding empirical adoption rate. The diagonal line indicates that the fitted adoption rate is the same as the empirical data. (A) demonstrates results in Washington and (B) demonstrates results in California.

Comments from Reviewer 3

Thank you to the authors for submitting this paper. I enjoyed reading it and I think it is an important contribution to the literature. I hope that the below comments will help the authors in making the paper as impactful as possible. The paper presented concerns about an alternate approach to modeling BEV adoption using the diffusion of innovations theory. The specific advance is the use of an augmented BDM which accounts for the social network among the population of potential adopters. The paper highlights a stark contrast between long-term predictions generated using the SocNet BDM and traditional BDM using the same data. Challenging the use of the BDM for EV adoption is very relevant as so much policy is based on it and it is probably too simple to be useful. This result is important and fitting for Nature. Method descriptions are sound and well-presented. The paper itself requires changes. In particular, the authors make many claims about the physical meaning of model parameters which are not justifiable. The parameters of the BDM are descriptive and fit to data - the authors do not provide or cite proof that the p -parameter is correlated to vehicle net cost, income, etc. but this assumption is manifested throughout the paper. The authors do not sufficiently justify that the SocNet model is a better reflection of reality than a geometry-based model but do claim this several times. The SocNet model introduces an additional parameter that must be either assumed or fit to data - the authors test three values for this parameter but do not justify why they chose these values.

Response: Thank you very much for your valuable feedback and suggestions. Our team sincerely appreciates your thoughtful review and the time you dedicated to helping us improve the manuscript. We have carefully considered your comments and, in the following text, have addressed them point-by-point.

1. That the SocNet BDM can produce such different values for the same input data is an interesting finding. However, in both the introduction and discussion the authors frame this result as the BDM majorly overpredicting the more realistic SocNet BDM. The problem is that the SocNet BDM is not obviously proven to be more realistic and the SocNet BDM produces a wide range of predictions depending on the gamma parameter. As written, this paper proves that a different set of assumptions than usually applied leads to very different conclusions, not that these conclusions are more accurate. “Interestingly, while both models fit current adoption data until 2022, when projecting into the future, models that omit spatial social networks will result in four times more adopters by 2050.” Could be reworded to be more direct, less biased, and even emphasized as this is the key finding. Suggestion: “Pre-2022 EV adoption has largely been driven by personal decisions rather than social interactions, a dynamic which cannot continue. Thus both models can be fit to pre-2022 data but the decision to include or omit the effects of social networks results in a factor of four divergence in predictions by 2050.” or similar.

Response: Thank you for bringing this important issue into discussion. We appreciate your insights and agree with you. As per your suggestion, we have revised the introduction and discussion accordingly. In Section 2.4, we have removed the argument about “SocNet BM uses

a more realistic social network,” as well as the statement: ”People in a more realistic social network are only influenced by their social contacts.”

We revised the manuscript accordingly.

- Section 1 Introduction: ~~Interestingly, while both models fit current adoption data until 2022 when projecting into the future, models that omit spatial social networks will result in four times more adopters by 2050.~~ Results reveal that both models can be calibrated to fit data up to 2022, but including or excluding social network effects leads to a ~~fourfold~~ threefold adoption rate difference in projections by 2050.
- Section 2.4 Divergence of PEVs Adoption Forecasts: This is because the SocNet BM uses a ~~more realistic and less dense~~ social network model while individuals are influenced by the whole population in the State/County BM. ~~In the State/County BM, individuals are influenced by the whole population while people in a more realistic social network are only influenced by their social contacts. Therefore, an adopter only influences a reduced number of people, slowing down the adoption process.~~
- Section 3 Discussion: Furthermore, this study demonstrates that the SocNet BM with the spatial social network as an extra ingredient allows more flexibility to capture the spatial autocorrelation of the empirical diffusion process.

2. This study uses 2022 and prior year data which is fine but the authors should note where there has been significant development by time of submission. Per CalTrans the BEV stock in California was 3.8% by the end of 2023 (<https://www.energy.ca.gov/data-reports/energy-almanac/zero-emission-vehicle-and-infrastructure-statistics-collection/light>) which crosses over into Early Adopter territory.

Response: Thank you for your constructive suggestion and for sharing the study on the ongoing BEV adoption trend. We have incorporated this insight into the discussion section.

We revised the manuscript accordingly.

- Section 3 Discussion: Last, this study uses data previous to 2023, but we note that the BEV stock in California was 3.8% by the end of 2023 [16], which crosses into early adopter territory. This allows us new opportunities to predict the diffusion in a more detailed way. The segmentation based on consumer characteristics alone is a valid method for forecasting PEV adoption [17]. However, consideration of the vehicle model is crucial to connect adoption with the power grid. Currently, due to limited data, we focus on PEVs without distinguishing between BEVs and PHEVs. With more data, it will become feasible to predict the adoption of BEVs and PHEVs separately and to investigate the distinct implications for the power grid related to BEVs versus PHEVs, along with consumer profiles and motivations.

3. Figure 1: C, F - confidence intervals are barely visible and the trend line is invisible where observations are thick. It would be better if the authors could somehow zoom in on California, perhaps by showing a “zoom-in” on the Bay Area.

Response: Thank you for your suggestion. We have revised Figures 1 (C, F) by updating the color scheme to make the trend lines more visible. Additionally, we have added a zoomed-in view of the California Bay Area in Figure 1 (E) to provide more detailed information.

We revised the manuscript accordingly.

– Section 2.1 Adoption Overview: We revised Fig. 8 as follows.

Figure 8: Overview of current PEV adoption. (A,D) State-level yearly total adoption rate (circle marker) and new adoption rate (square marker) in Washington (blue) and California (orange) from 2010 to 2022. (B,E) Maps of county-level cumulative adoption rate in Washington (blue) and California (orange) in 2022. A zoom-in adoption rate map in Bay Area is attached with (E). (C,F) Correlation between the adoption rate and the corresponding median household income of all census tracts in Washington (blue) and California (orange) in 2022. The black regression line given the adoption rate and the corresponding median household income, ~~as well as the 95% confidence interval for the regression,~~ are shown with points.

4. Figure 2: B, D - The legend is cut off by the bars in both cases.

Response: Thank you for your suggestion. We have revised Figures 2 (B) and (D) to ensure that the legends are no longer cut off by the bars.

We revised the manuscript accordingly.

– Section 2.2 Calibration of PEVs Diffusion Models: We revised Fig. 9 as follows.

Figure 9: Comparison of State/County BM and SocNet BM's performance. (A, B) State-level fitted adoption curves of State BM (cross), County BM (star), SocNet BM (circle), and data (square) for low (light blue/yellow), middle (blue/orange), and high (dark blue/red) income groups in Washington and California from 2010 to 2022. Solid lines with markers show the median (50th percentile) adoption rate of income groups in states; the shaded bands show the range from the 5th to 95th percentile, highlighting the uncertainty. (C, D) Distribution of tract level adoption rate error of State BM (cross), County BM (star), and SocNet BM (circle) in Washington (blue) and California (orange) in 2022. (E, F, G) County-level adoption rates of State/County BM and SocNet BM in the most populated 20 15 counties in Washington (blue) and California (orange) in 2022. The box shows the quartiles of the county-level adoption rates of SocNet BM while the whiskers extend to show the rest of the distribution. Cross markers show the adoption rate of State BM, star markers show the adoption rate of County BM, and square markers show the empirical data.

5. General comment is that the size and color scheme adopted in Figures 1 and 2 do not help legibility. The authors could choose colors that stand out better against a white background, change to a darker background, increase figure size, add grid lines, etc.

Response: Thank you for your remark. We have adjusted the size and color scheme in Figures 1 and 2 by increasing the figure size and selecting colors that stand out more clearly against the white background.

We revised the manuscript accordingly.

– Section 2.1 Adoption Overview: We revised Fig. 10 as follows.

Figure 10: Overview of current PEV adoption. (A,D) State-level yearly total adoption rate (circle marker) and new adoption rate (square marker) in Washington (blue) and California (orange) from 2010 to 2022. (B,E) Maps of county-level cumulative adoption rate in Washington (blue) and California (orange) in 2022. A zoom-in adoption rate map in Bay Area is attached with (E). (C,F) Correlation between the adoption rate and the corresponding median household income of all census tracts in Washington (blue) and California (orange) in 2022. The black regression line given the adoption rate and the corresponding median household income, as well as the 95% confidence interval for the regression, are shown with points.

– Section 2.2 Calibration of PEVs Diffusion Models: We revised Fig. 11 as follows.

Figure 11: Comparison of State/County BM and SocNet BM's performance. (A, B) State-level fitted adoption curves of State BM (cross), County BM (star), SocNet BM (circle), and data (square) for low (light blue/yellow), middle (blue/orange), and high (dark blue/red) income groups in Washington and California from 2010 to 2022. Solid lines with markers show the median (50th percentile) adoption rate of income groups in states; the shaded bands show the range from the 5th to 95th percentile, highlighting the uncertainty. (C, D) Distribution of tract level adoption rate error of State BM (cross), County BM (star), and SocNet BM (circle) in Washington (blue) and California (orange) in 2022. (E, F, G) County-level adoption rates of State/County BM and SocNet BM in the most populated 20 15 counties in Washington (blue) and California (orange) in 2022. The box shows the quartiles of the county-level adoption rates of SocNet BM while the whiskers extend to show the rest of the distribution. Cross markers show the adoption rate of State BM, star markers show the adoption rate of County BM, and square markers show the empirical data.

6. Table 1 and two paragraphs below: The authors make a great deal of interpreting fitted BDM coefficients. Specifically they give the reader the impression that these coefficients are fundamental behavioral characteristics of the relevant populations. They are not - they are descriptive. Specifically, the authors do not prove that a higher p -coefficient is due to “higher household incomes, they know more financial incentives policies, or they have more access to charging stations”. Please either provide evidence to back this claim or remove it.

Response: Thank you very much for your remark. We agree with you that the coefficients of BM are descriptive. Therefore, we removed this claim from the manuscript.

We revised the manuscript accordingly.

- Section 2.3 Parameters of PEVs Diffusion Models: We show the distribution of obtained parameters for State/County BM and SocNet BM for the three socio-demographic groups in Washington and California in Fig. 3 and Table 1. and attach median values of these parameters in Supplementary Information (see Supplementary Information Note S3). Fig. 3 (A) shows that in Washington, the high-income group has the highest p and the low-income group has the lowest p for three models. This means that the high-income group is more likely to adopt PEVs due to intrinsic factors. For example, they have higher household incomes, they know more financial incentives policies, or they have more access to charging stations compared with the middle-income group and low-income group. The difference between County BM and SocNet BM is that the parameters of SocNet BM have more variance. This variance is brought by fitting with limited data, and also the probabilistic nature of the agent-based simulation process. Fig. 3 (B) shows that the q for State/County BM are similar across three groups in Washington. The q for the low-income group is slightly higher, which means people with a lower income are more likely to adopt EVs because of extrinsic factors. Unlike with State/County BM, under SocNet BM the middle-income group has a higher median q , indicating that people of the middle-income group are more likely to adopt due to social network effects. Fig. 10 (C) shows that in California, the high-income group has the highest p and the low-income group has the lowest p . This implies that similar to Washington, people of the high-income group in California are more likely to adopt EVs due to intrinsic factors. Similar to Washington, the high-income group in California has the highest p and the low-income group has the lowest p as shown in Fig. 3 (C). Compared with Washington, California has higher p for the three income groups at the state level. This implies that people in California have a higher motivation to adopt PEVs than people in Washington. Fig. 3 (D) shows that in California, the q of the State/County BM and SocNet BM are similar across three groups. The difference between State/County BM and SocNet BM is that the fitted parameters of SocNet BM have more variance. This variance is brought by fitting with limited data, and also the probabilistic nature of the agent-based simulation process. in California is similar to Washington in terms of having q for the low-income group being the highest. This means that for both states, the low-income population is more likely to adopt PEVs because of extrinsic motivations. Unlike in Washington, the SocNet BM in California has the middle-income group with the lowest median q , indicating that they are less influenced by their social contacts when making decisions than the other groups.

7. Second Paragraph of 2.4. The authors express an opinion here (and in the introduction) that the SocNet model has a more realistic interpretation of interaction and influence but insufficient evidence is provided to justify this claim. Specifically the authors argue “people in a more realistic social network are only influenced by their social contacts”. Without contrary evidence, it is not ridiculous to think that consumers may be influenced by seeing EVs around even if they don’t know the owners. With the evidence presented the authors must state that the SocNet model is based on different assumptions without claiming that these are more realistic.

Response: Thank you for bringing this important issue into the discussion. We agree with you that although SocNet BM introduces the social network effects in the modeling, it is another model we do not have evidence that is more realistic. Therefore, we revise the second Paragraph of 2.4, the introduction, and the discussion.

We revised the manuscript accordingly.

- Section 1 Introduction: ~~Interestingly, while both models fit current adoption data until 2022 when projecting into the future, models that omit spatial social networks will result in four times more adopters by 2050.~~ Results reveal that both models can be calibrated to fit data up to 2022, but including or excluding social network effects leads to a ~~fourfold~~ threefold difference in adoption rate projections by 2050.
- Section 2.4 Divergence of PEVs Adoption Forecasts: This is because the SocNet BM uses a ~~more realistic and less dense~~ social network model while individuals are influenced by the whole population in the State/County BM. ~~In the State/County BM, individuals are influenced by the whole population while people in a more realistic social network are only influenced by their social contacts. Therefore, an adopter only influences a reduced number of people, slowing down the adoption process.~~
- Section 3 Discussion: Furthermore, this study demonstrates that the SocNet BM with the spatial social network as an extra ingredient allows more flexibility to capture the spatial autocorrelation of the empirical diffusion process.

8. Section 2.5: Insight into a physical manifestation of the effects of gamma parameter would be useful. For example - could the authors plot distributions of valency and edge length for different gamma? - Could the authors explain the selection of 1, 3, 10 and which they think is most realistic/why?

Response: Thank you for your suggestion. To provide a more comprehensive view of the social network we created and justify the γ value selection of 1, 3, and 10, we add one note in Supplementary Information.

We revised the manuscript accordingly.

- Section 4.3.1 Social Network Construction: To include preferential attachment and distance selection, we use $K(k) = k + 1$ and $D(d) = d^\gamma$, where $\gamma = 1, 3, 10$ (see Supplementary Information Note S9 for the rationale behind the selection of γ values).
- Note S9 Justification of Values of γ s: We calculate clustering coefficient [57], assortativity [58], the distribution of degree, and distribution of the geographical distance between nodes in Los

Angeles. Table 4 and Fig. 12 show the overview of social network characteristics with different γ . We find that higher γ leads to higher clustering coefficients, assortativity, $P(k)$ exponents, and $P(r)$ exponents.

Previous research finds for the social network inferred from blogs [59], location-based social networks [60] [61] or mobile phone data [62] [63] [64], the distribution of geographical distance between nodes follows a power law, with exponents between -1 and -2. Therefore, we explore social networks with $\gamma = 1, 3, 10$, which leads exponents between 0 to -3. Also, we set $\gamma = 10$ for the default social network for analysis as it leads to a clustering coefficient closer to empirical values in social networks.

γ	Clustering Coefficient	Assortativity	P(k) Power Law Fitted Exponent	P(r) Power Law Fitted Exponent
1	0.0001	0.07	-3	-0.3
3	0.01	0.65	-3	-2.1
10	0.04	0.96	-3	-3.0

Table 4: Social Network Characteristics with Different γ

Figure 12: Distance distribution $P(r)$ and Degree distribution $P(k)$ of social network with $\gamma = 1, 3, 10$. (A) Distance distribution of various social network structures. The solid line indicates the data distribution and the dashed line with the same color indicates the fitted power law distribution. (B) Degree distribution of various social network structures. The solid line indicates the data distribution and the grey dashed line indicates the fitted power law distribution.

9. Section 2.5: There is a lot of information in this section but the most important point is that spatial autocorrelation seems to line up best with the $\gamma = 1$ data. The authors could both emphasize this and its implications and reduce the rest of the discussion.

Response: Thank you for your remark. In response to your comment on the statement "SocNet BM with $\gamma = 1$ best reflects the observed spatial autocorrelation of PEV adoption," we extended our analysis to demonstrate Moran's I for additional counties. Our findings show that the optimal γ value varies across counties. Based on these results, we conclude that "the SocNet BM with spatial social network allows more flexibility to capture the spatial autocorrelation of the empirical diffusion process". We emphasize this conclusion and reduce the rest of the discussion in the manuscript.

We revised the manuscript accordingly.

- Section 2.5 Effects of Network Structure on Spatial Diffusion: With networks with $\gamma = 1, 3, 10$, calibrated models show diverse adoption predictions in spatial characteristics. Fig. 5 (A) shows the adoption rates with corresponding Moran's I statistics under $\gamma = 1, 3, 10$ in Los Angeles. When $\gamma = 1$, Moran's I statistics first increase and then decrease with growing adoption rates. In contrast, with $\gamma = 3$ and 10, Moran's I statistics monotonically increase with growing adoption rates. Fig. 5 (B) shows that Moran's I statistics of State/County BM are much higher than the data. ~~This is because, f~~ For State/County BM, we only know the state/county-level aggregated adoption rate. To obtain census tract-level adoption rates, we assign adopters proportionally to the population in each census tract (see Methodology 4.2). ~~Then in one county, these two models only have three unique adoption rates (corresponding to the three income groups). This results in more neighbors with the same adoption rates, which increases spatial autocorrelation. We also explore~~ In summary, Fig. 5 (A-B) demonstrate how different network structures influence the adoption by changing the γ of networks in the SocNet BM. ~~The case with $\gamma = 1$ has lower Moran's I statistics than the case with $\gamma = 3$ because the former has more distant links so adoption diffusion is more balanced. In contrast, the latter has more links with neighbors so the adoption diffusion is more clustered. Interestingly, for $\gamma = 10$, Moran's I statistics decrease compared with the case with $\gamma = 3$. That means increasing the number of close connections doesn't increase Moran's I statistics. This is because of the slower adoption caused by increasing γ : with less distant links, agents far away from initial adopters will not be able to be influenced by social network effects. With a slower adoption diffusion, the cluster area is smaller and thus decreases spatial autocorrelation. As different social network structures lead to different spatial patterns of adoption, we can infer social network structure based on empirical adoption data. As different social network structures lead to different spatial patterns of adoption, we can infer the best social network model to represent the adoption based on the spatial auto-correlation of the data. Specifically, in Los Angeles, $\gamma = 10$ appears to better characterize the social network associated with PEV adoption (see Supplementary Information Note S6 for results in other counties). Overall, the SocNet BM with spatial social network allows more flexibility to capture the spatial autocorrelation of the empirical diffusion. Fig. 5 (B) shows the adoption rates with corresponding Moran's I statistics under $\gamma = 1, 3, 10$. When $\gamma = 1$, Moran's I statistics first increase and then decrease with growing adoption rates. When adoption rates are below 10%, Moran's I statistics increase due to few initial adopters creating small clusters. Once reached a 10% adoption rate, $\gamma = 1$ leads to a more balanced adoption diffusion in Los Angeles (indicated by decreasing Moran's I statistics). In contrast, with $\gamma = 3, 10$, Moran's I statistics monotonically increase with growing adoption rates. This is because higher γ~~

leads to a network with more closer links. Thus the adoption diffusion is more unbalanced, i.e., in some areas the diffusion is fast; in other places it's slow. This unbalanced diffusion causes (1) higher spatial autocorrelation when the adoption rate is low, as well as (2) a slower diffusion than in the case of $\gamma = 1$. This slower diffusion also makes the $\gamma = 10$ case have lower spatial autocorrelation than the case of $\gamma = 3$. Besides, we also find that when $\gamma = 10$, Moran's I statistics have a higher variance than the case with $\gamma = 3$. This indicates when the adoption rate is similar, the $\gamma = 10$ case is more likely to have extremely higher or lower spatial autocorrelation, depending on whether the distant agents are activated.

- **Note S6 Urban Layout and Diffusion Process:** In terms of social structure, Fig. 13 (A-C) display the Moran's I statistics for State/County BM and SocNet BM across three counties in Washington, while Fig. 13 (D-F) show the same in California. In California and Washington's most populous counties, specifically King and Los Angeles, we can identify a social network structure parameter (γ) that closely replicates the observed Moran's I statistics. Nonetheless, counties such as Santa Clara and Pierce have no γ value that can accurately replicate the empirical Moran's I statistics.

Figure 13: Moran's I statistics for State/County BM and SocNet BM in 6 counties in Washington and California. For SocNet BM, we use the network with three γ s, i.e., 1,3,10. The fitting process is repeated for each γ and the median of predictions is reported. Solid lines with circle markers show the median (50th percentile) Moran's I statistics from SocNet BM. Dashed lines with cross markers and star markers show the Moran's I statistics from State/County BM; Squares show the Moran's I statistics calculated from empirical data on tract level.

10. Figure 5: Panel A - what data did the authors use to compute the I statistic in the series labeled "data" (gray squares)? Is this tract level?

Response: Thank you for your careful reading, and we apologize for any confusion. In Fig. 5 (A), we use empirical tract-level adoption data from 2010 to 2022 to calculate the Moran's I statistics.

We revised the manuscript accordingly.

- Section 2.5 Effects of Network Structure on Spatial Diffusion: Figure 5: Squares show the Moran’s I statistics calculated from empirical data on tract level.

11. Section 2.6: This section is technically sound but all previous comments on the link between the model and reality are relevant. Also what do the authors think the “influencer” strategy would entail? Would a company distribute personalized incentives to individuals based on social media following?

Response: Regarding the “link between the model and reality”, we revised the discussion about the implications of Fig. 7 and removed claims about connections between the model and implications in reality.

The essence of an influencer strategy is to strategically choose non-adopters based on their social connections and activate them as adopters so as to influence the PEV diffusion in social networks. Examples of influencer strategies include distributing personalized incentives to influencers, such as sending them PEVs or offering free PEV leases to them for certain periods. These influencers could then share their driving experiences or highlight the advantages of PEVs with their followers to promote adoption.

We revised the manuscript accordingly.

- Section 2.6 Promoting PEVs Adoption under Uncertainty: Examples include sending PEVs or offering free leases of PEVs to influencers for a certain period, so as to let them share driving experience or advantages of PEVs to their followers.
- Section 2.6 Promoting PEVs Adoption under Uncertainty: Fig. 7 (B-D) compares innovation and imitation-related adoption boosting strategies, applied to the low-income group and for three pairs of \hat{p}_{low} and \hat{q}_{low} that fit current data. We represent the “innovation-related” strategy as an increase of \hat{p}_{low} in a fixed factor, which modifies the global propensity of innovation adoption within the low-income group. We represent the “imitation-based” strategy by setting the first n agents with the highest degree as adopters. Each figure shows years to critical adoption rates (16%) resulting from the innovation-related and the imitation-related strategies for each pair $(\hat{p}_{low}, \hat{q}_{low})$. Fig. 7 (B) illustrates that with a relatively low fitted \hat{p}_{low} and high \hat{q}_{low} , an imitation-related approach with more influencers can effectively boost adoption, whereas increasing the fixed factor (innovation-related strategy) does not have much effect. Conversely, Fig. 7 (D) depicts a scenario with a relatively high \hat{p}_{low} and low \hat{q}_{low} , where increasing fixed factors significantly enhances adoption, but adding more influencers has a negligible effect. Fig. 7 (C) presents a middle ground, where combining both strategies achieves the highest adoption increase. It is crucial to note that the scenario where imitation-related strategies have a more significant impact does not imply that policies and actions seeking a global effect do not matter. On the contrary, initiatives such as consumer incentives or the development of charging infrastructure are essential for facilitating adoption in low-income areas or regions with limited charging stations. However, which are the best actions to increase the innovator rate is not clear. For example, Slowik *et al.* [2] assesses the PEV market in the 50 most populous metropolitan areas in the United States. They demonstrate that for areas with high PEV penetration, there is heterogeneity in the availability of policies, charging infrastructure, consumer incentives, and vehicle models. Altogether, Fig. 7 (B-D) shows that with different \hat{p}_{low} and \hat{q}_{low} effectively fitting the data, the year to reach critical adoption for different combinations of imitation-related strategy and innovation-related strategies varies as the contour of the year achieving critical adoption ranges from horizontally displayed and vertically displayed.

12. Discussion: 20250 should be 2050.

Response: Thank you for your careful reading. We have revised the sentence.

We revised the manuscript accordingly.

- Section 3 Discussion: Our comparison of projections for future EV adopters reveals that by introducing social network, the number of adopters by ~~20250~~ 2050 is 25% of benchmark projections without network structure.

13. Discussion: The authors should consider what the key takeaways of this study are. In the opinion of the reviewer, the strongest points are that (1) A more sophisticated diffusion model, fitted equally well to the same data, can produce results drastically different than traditional BDM implying substantial risk (2) The SocNet BDM with low gamma best reflects the observed spatial autocorrelation of BEV adoption.

Response: Thank you for your constructive suggestions. We agree with you that our first key message is that "a more sophisticated diffusion model, fitted equally well to the same data, can produce results drastically different from the traditional Bass model, implying substantial risk". Regarding the second message, we have demonstrated Moran's I for other counties in Supplementary Information and found that the γ reflecting empirical spatial autocorrelation varies across counties. Therefore, our revised second takeaway message now reads: "SocNet BM with the spatial social network as an extra ingredient allows more flexibility to capture the spatial autocorrelation of the empirical diffusion process."

We revised the manuscript accordingly.

- Section 3 Discussion: Our comparison of projections for future PEV adopters reveals that ~~by introducing social network, the number of adopters by 20250 is 25% of benchmark projections without network structure. Besides, measuring spatial autocorrelation through Moran's I statistics reveals that the model considering spatial social network characterizes the spatial pattern better than the BM. Moreover, we find social network structures influence the geographical distribution of future EV adopters. Specifically, more distant connections in social networks lead to balanced diffusion, while more close connections lead to clustered adoption patterns and slower adoption, highlighting the need to consider various network structures when characterizing the future geographical distribution of EV adopters.~~ a more sophisticated diffusion model, fitted equally well to the same data, can produce results drastically different than the traditional BMs. The divergence of forecasts also indicates that it's important to consider the complexity of the current model design. While the more parameters in these models facilitate a closer fit to the available data, there is an inherent risk of overfitting. ~~A careful balance is needed to ensure the models are not excessively intricate while effectively capturing key trends.~~ Furthermore, this study demonstrates that the SocNet BM with the spatial social network as an extra ingredient allows more flexibility to capture the spatial autocorrelation of the empirical diffusion process.

14. The code is not publicly available at the address provided. I copied the URL and got an error 404. I went to the HuMNet lab git page and did not find it among the public repositories. Thus I did not review the code.

Response: Thank you for your careful reading, and we apologize for the inconvenience caused by the inaccessible link. Please refer to the following updated link for the code and data: github.com/humnetlab/Planning-the-Electric-Vehicle-Transition-by-Integrating-Spatial-Information-and-Social-Networks.

We revised the manuscript accordingly.

- Code and Data Availability: The code and data used for the analysis presented in this paper are available at GitHub: github.com/humnetlab/Planning-the-Electric-Vehicle-Transition-by-Integrating-Spatial-Information-and-Social-Networks. Please contact M.C.G (martag@berkeley.edu) and J. W. (jmwu@berkeley.edu) for any questions.

Comments from Reviewer 4

Response: Thank you for your feedback. Our team acknowledges the assessment. In the following text, we have addressed your comments in a point-by-point manner.

1. The code link is not valid.

Response: Thank you for your careful reading, and we apologize for the inconvenience caused by the inaccessible link. Please refer to the following updated link for the code and data: github.com/humnetlab/Planning-the-Electric-Vehicle-Transition-by-Integrating-Spatial-Information-and-Social-Networks.

We revised the manuscript accordingly.

- Code and Data Availability: The code and data used for the analysis presented in this paper are available at GitHub: github.com/humnetlab/Planning-the-Electric-Vehicle-Transition-by-Integrating-Spatial-Information-and-Social-Networks. Please contact M.C.G (martag@berkeley.edu) and J. W. (jmwu@berkeley.edu) for any questions.

References

- [1] C. Münzel, P. Plötz, F. Sprei, and T. Gnann, “How large is the effect of financial incentives on electric vehicle sales?—a global review and european analysis,” *Energy Economics*, vol. 84, p. 104493, 2019.
- [2] P. Slowik and N. Lutsey, “Expanding the electric vehicle market in us cities,” ICCT Washington, DC, USA, 2017.
- [3] J. H. Lee, S. J. Hardman, and G. Tal, “Who is buying electric vehicles in california? characterising early adopter heterogeneity and forecasting market diffusion,” *Energy Research & Social Science*, vol. 55, pp. 218–226, 2019.
- [4] V. Singh, V. Singh, and S. Vaibhav, “A review and simple meta-analysis of factors influencing adoption of electric vehicles,” *Transportation Research Part D: Transport and Environment*, vol. 86, p. 102436, 2020.
- [5] California Energy Commission, “Home charging access in california.” <https://www.energy.ca.gov/publications/2022/home-charging-access-california>, 2022. Accessed on May 1, 2022.
- [6] A. Chakraborty, R. R. Kumar, and K. Bhaskar, “A game-theoretic approach for electric vehicle adoption and policy decisions under different market structures,” *Journal of the Operational Research Society*, vol. 72, no. 3, pp. 594–611, 2021.
- [7] U. Illmann and J. Kluge, “Public charging infrastructure and the market diffusion of electric vehicles,” *Transportation Research Part D: Transport and Environment*, vol. 86, p. 102413, 2020.
- [8] Y. Shi, Z. Wei, M. Shahbaz, and Y. Zeng, “Exploring the dynamics of low-carbon technology diffusion among enterprises: An evolutionary game model on a two-level heterogeneous social network,” *Energy Economics*, vol. 101, p. 105399, 2021.
- [9] R. Debnath, R. Bardhan, D. M. Reiner, and J. Miller, “Political, economic, social, technological, legal and environmental dimensions of electric vehicle adoption in the united states: A social-media interaction analysis,” *Renewable and Sustainable Energy Reviews*, vol. 152, p. 111707, 2021.
- [10] F. Mandys, “Electric vehicles and consumer choices,” *Renewable and Sustainable Energy Reviews*, vol. 142, p. 110874, 2021.
- [11] X. Zhao, X. Li, Z. Zhao, and T. Luo, “Media attention and electric vehicle adoption: Evidence from 275 cities in china,” *Transportation Research Part A: Policy and Practice*, vol. 190, p. 104269, 2024.
- [12] F. Nazari, A. K. Mohammadian, and T. Stephens, “Modeling electric vehicle adoption considering a latent travel pattern construct and charging infrastructure,” *Transportation Research Part D: Transport and Environment*, vol. 72, pp. 65–82, 2019.
- [13] A. M. Brockway, J. Conde, and D. Callaway, “Inequitable access to distributed energy resources due to grid infrastructure limits in california,” *Nature Energy*, vol. 6, no. 9, pp. 892–903, 2021.
- [14] Y. Yuan, Y. Zhou, Z. Lin, and K. Jin, “Prediction of pev adoption with agent-based parameterized bass network diffusion model,” *arXiv preprint arXiv:2303.15313*, 2023.

- [15] E. Kiesling, *Planning the market introduction of new products: An agent-based simulation of innovation diffusion*. na, 2011.
- [16] California Energy Commission, “Light-duty vehicle population in california.” <https://www.energy.ca.gov/data-reports/energy-almanac/zero-emission-vehicle-and-infrastructure-statistics-collection/light>, 2024.
- [17] T. Gnann, T. S. Stephens, Z. Lin, P. Plötz, C. Liu, and J. Brokate, “What drives the market for plug-in electric vehicles?-a review of international pev market diffusion models,” *Renewable and Sustainable Energy Reviews*, vol. 93, pp. 158–164, 2018.
- [18] F. M. Bass, “A new product growth for model consumer durables,” *Management science*, vol. 15, no. 5, pp. 215–227, 1969.
- [19] J. Liu, C. Zhuge, J. H. C. G. Tang, M. Meng, and J. Zhang, “A spatial agent-based joint model of electric vehicle and vehicle-to-grid adoption: A case of beijing,” *Applied Energy*, vol. 310, p. 118581, 2022.
- [20] Q. Zhang, J. Liu, K. Yang, B. Liu, and G. Wang, “Market adoption simulation of electric vehicle based on social network model considering nudge policies,” *Energy*, vol. 259, p. 124984, 2022.
- [21] J. Jansson, T. Pettersson, A. Mannberg, R. Brännlund, and U. Lindgren, “Adoption of alternative fuel vehicles: Influence from neighbors, family and coworkers,” *Transportation Research Part D: Transport and Environment*, vol. 54, pp. 61–73, 2017.
- [22] D. Chakraborty, D. S. Bunch, D. Brownstone, B. Xu, and G. Tal, “Plug-in electric vehicle diffusion in california: Role of exposure to new technology at home and work,” *Transportation Research Part A: Policy and Practice*, vol. 156, pp. 133–151, 2022.
- [23] U.S. Energy Information Administration, “U.s. share of electric and hybrid vehicle sales reached a record in the third quarter.” <https://www.eia.gov/todayinenergy/detail.php?id=63904>, 2024.
- [24] Center for Sustainable Energy, “California air resources board clean vehicle rebate project.” <https://cleanvehiclerebate.org/cvrp-rebate-map>, 2023. Retrieved on March, 2023.
- [25] Washington State Department of Licensing, “Electric vehicle title and registration activity.” https://data.wa.gov/Transportation/Electric-Vehicle-Title-and-Registration-Activity/rpr4-cgyd/about_data, 2023. Retrieved on March, 2023.
- [26] U.S. Department of Transportation, “2022 national household travel survey.” <https://nhts.ornl.gov>, 2022.
- [27] H. Byun, J. Shin, and C.-Y. Lee, “Using a discrete choice experiment to predict the penetration possibility of environmentally friendly vehicles,” *Energy*, vol. 144, pp. 312–321, 2018.
- [28] Z. Duan, B. Gutierrez, and L. Wang, “Forecasting plug-in electric vehicle sales and the diurnal recharging load curve,” *IEEE Transactions on Smart Grid*, vol. 5, no. 1, pp. 527–535, 2014.
- [29] C. R. Forsythe, K. T. Gillingham, J. J. Michalek, and K. S. Whitefoot, “Technology advancement is driving electric vehicle adoption,” *Proceedings of the National Academy of Sciences*, vol. 120, no. 23, p. e2219396120, 2023.

- [30] J. Wu, S. Powell, Y. Xu, R. Rajagopal, and M. C. Gonzalez, “Planning charging stations for 2050 to support flexible electric vehicle demand considering individual mobility patterns,” *Cell Reports Sustainability*, vol. 1, no. 1, 2024.
- [31] T. A. Becker, I. Sidhu, and B. Tenderich, “Electric vehicles in the united states: a new model with forecasts to 2030,” *Center for Entrepreneurship and Technology, University of California, Berkeley*, vol. 24, pp. 1–32, 2009.
- [32] A. Jenn, G. Tal, and L. Fulton, “A multi-model approach to generating international electric vehicle future adoption scenarios,” in *Proceedings of the EVS30: 30th International Electric Vehicle Symposium & Exhibition, Stuttgart, Germany*, pp. 9–11, 2017.
- [33] T. Wang, Z. Jiang, B. Zhao, Y. Gu, K.-N. Liou, N. Kalandiyur, D. Zhang, and Y. Zhu, “Health co-benefits of achieving sustainable net-zero greenhouse gas emissions in california,” *Nature Sustainability*, vol. 3, no. 8, pp. 597–605, 2020.
- [34] Q. Yu, B. Y. He, J. Ma, and Y. Zhu, “California’s zero-emission vehicle adoption brings air quality benefits yet equity gaps persist,” *Nature Communications*, vol. 14, no. 1, p. 7798, 2023.
- [35] Alternative Fuels Data Center, “Electric vehicle benefits and considerations.” <https://afdc.energy.gov/fuels/electricity-benefits>, 2022.
- [36] California Air Resources Board, “California moves to accelerate to 100% new zero-emission vehicle sales by 2035.” <https://ww2.arb.ca.gov/news/california-moves-accelerate-100-new-zero-emission-vehicle-sales-2035>, 2022.
- [37] Washington State Department of Ecology, “Washington sets path to phase out gas vehicles by 2035.” <https://ecology.wa.gov/about-us/who-we-are/news/2022/sept-7-clean-vehicles-public-comment>, 2022.
- [38] Council of the People’s Republic of China, “Development plan for the new energy vehicle industry (2021–2035).” https://www.gov.cn/zhengce/content/2020-11/02/content_5556716.htm, 2020.
- [39] Department for Transport U.K, “Transitioning to zero-emission cars and vans: 2035 delivery plan. gov.uk 57.” <https://www.gov.uk/government/publications/transitioning-to-zero-emission-cars-and-vans-2035-delivery-plan>, 2021.
- [40] Council of the European Union, “First ‘fit for 55’ proposal agreed: the eu strengthens targets for co2 emissions for new cars and vans.” <https://www.consilium.europa.eu/en/press/press-releases/2022/10/27/first-fit-for-55-proposal-agreed-the-eu-strengthens-targets-for-co2-emissions-for-new-cars-and-vans/>, 2022.
- [41] Amanda Myers, “4 u.s. electric vehicle trends to watch in 2019.” <https://www.forbes.com/sites/energyinnovation/2019/01/02/4-u-s-electric-vehicle-trends-to-watch-in-2019/#412908785a3c>, 2019.
- [42] S. Powell, G. V. Cezar, L. Min, I. M. Azevedo, and R. Rajagopal, “Charging infrastructure access and operation to reduce the grid impacts of deep electric vehicle adoption,” *Nature Energy*, vol. 7, no. 10, pp. 932–945, 2022.

- [43] R. Gupta, A. Pena-Bello, K. N. Streicher, C. Roduner, Y. Farhat, D. Thöni, M. K. Patel, and D. Parra, “Spatial analysis of distribution grid capacity and costs to enable massive deployment of pv, electric mobility and electric heating,” *Applied Energy*, vol. 287, p. 116504, 2021.
- [44] R. J. Javid and A. Nejat, “A comprehensive model of regional electric vehicle adoption and penetration,” *Transport Policy*, vol. 54, pp. 30–42, 2017.
- [45] S. Wee, M. Coffman, and S. Allen, “Ev driver characteristics: Evidence from hawaii,” *Transport Policy*, vol. 87, pp. 33–40, 2020.
- [46] M. Muratori, “Impact of uncoordinated plug-in electric vehicle charging on residential power demand,” *Nature Energy*, vol. 3, no. 3, pp. 193–201, 2018.
- [47] C. Crozier, T. Morstyn, and M. McCulloch, “The opportunity for smart charging to mitigate the impact of electric vehicles on transmission and distribution systems,” *Applied Energy*, vol. 268, p. 114973, 2020.
- [48] S. Powell, E. C. Kara, R. Sevlian, G. V. Cezar, S. Kiliccote, and R. Rajagopal, “Controlled workplace charging of electric vehicles: The impact of rate schedules on transformer aging,” *Applied Energy*, vol. 276, p. 115352, 2020.
- [49] E. Storage, “Impacts of electrochemical utility-scale battery energy storage systems on the bulk power system,” *NERC: Atlanta, GA, USA*, 2021.
- [50] J. Zhang, J. Jorgenson, T. Markel, and K. Walkowicz, “Value to the grid from managed charging based on california’s high renewables study,” *IEEE Transactions on Power Systems*, vol. 34, no. 2, pp. 831–840, 2018.
- [51] C. Liu, K. Chau, D. Wu, and S. Gao, “Opportunities and challenges of vehicle-to-home, vehicle-to-vehicle, and vehicle-to-grid technologies,” *Proceedings of the IEEE*, vol. 101, no. 11, pp. 2409–2427, 2013.
- [52] S. Khan, H. Maoh, and T. Dimatulac, “The demand for electrification in canadian fleets: A latent class modeling approach,” *Transportation Research Part D: Transport and Environment*, vol. 90, p. 102653, 2021.
- [53] G. Z. De Rubens, L. Noel, J. Kester, and B. K. Sovacool, “The market case for electric mobility: Investigating electric vehicle business models for mass adoption,” *Energy*, vol. 194, p. 116841, 2020.
- [54] G. Wang, K. Makino, A. Harmandayan, and X. Wu, “Eco-driving behaviors of electric vehicle users: A survey study,” *Transportation research part D: transport and environment*, vol. 78, p. 102188, 2020.
- [55] McKinsey & Company, “Mckinsey mobility consumer pulse.” https://executivedigest.sapo.pt/wp-content/uploads/2024/06/Mobility-Consumer-Pulse-2024_Overview.pdf, 2024.
- [56] US Census Bureau, “Census bureau data.” <https://data.census.gov>, 2023. Retrieved on March, 2023.
- [57] J. Saramäki, M. Kivelä, J.-P. Onnela, K. Kaski, and J. Kertesz, “Generalizations of the clustering coefficient to weighted complex networks,” *Physical Review E*, vol. 75, no. 2, p. 027105, 2007.

- [58] M. E. Newman, “Mixing patterns in networks,” *Physical review E*, vol. 67, no. 2, p. 026126, 2003.
- [59] D. Liben-Nowell, J. Novak, R. Kumar, P. Raghavan, and A. Tomkins, “Geographic routing in social networks,” *Proceedings of the National Academy of Sciences*, vol. 102, no. 33, pp. 11623–11628, 2005.
- [60] S. Scellato, A. Noulas, and C. Mascolo, “Exploiting place features in link prediction on location-based social networks,” in *Proceedings of the 17th ACM SIGKDD international conference on Knowledge discovery and data mining*, pp. 1046–1054, 2011.
- [61] E. Cho, S. A. Myers, and J. Leskovec, “Friendship and mobility: user movement in location-based social networks,” in *Proceedings of the 17th ACM SIGKDD international conference on Knowledge discovery and data mining*, pp. 1082–1090, 2011.
- [62] R. Lambiotte, V. D. Blondel, C. De Kerchove, E. Huens, C. Prieur, Z. Smoreda, and P. Van Dooren, “Geographical dispersal of mobile communication networks,” *Physica A: Statistical Mechanics and its Applications*, vol. 387, no. 21, pp. 5317–5325, 2008.
- [63] J.-P. Onnela, J. Saramäki, J. Hyvönen, G. Szabó, D. Lazer, K. Kaski, J. Kertész, and A.-L. Barabási, “Structure and tie strengths in mobile communication networks,” *Proceedings of the national academy of sciences*, vol. 104, no. 18, pp. 7332–7336, 2007.
- [64] C. Herrera-Yagüe, C. M. Schneider, T. Couronne, Z. Smoreda, R. M. Benito, P. J. Zufria, and M. C. González, “The anatomy of urban social networks and its implications in the searchability problem,” *Scientific reports*, vol. 5, no. 1, p. 10265, 2015.

Response to Reviewers:
“Planning the Electric Vehicle Transition by Integrating Spatial
Information and Social Networks”

Jiaman Wu, Ariel Salgado and Marta C. Gonzalez

September 25, 2025

Comments from Reviewer 2

The authors have done a great job to address reviewer comments. We do not have further concerns and support accepting this manuscript.

Response: We sincerely thank you for the encouraging comments and positive evaluation. Your constructive suggestions during the review process have greatly helped us improve the manuscript.

Comments from Reviewer 3

All of my comments have been well addressed, thanks! The code is servicable for validating results but is not in the format of a generally usable package. The code is largely not commented but descriptive variable names make it readable. Nevertheless, I would instruct a student to use the code as inspiration rather than attempt use the code itself. It would be nice if this was a PyPI package or, at least, structured like one. This would include the dependency management piece so users wouldn't have to set up an env manually.

Response: We are pleased that our revisions have satisfactorily addressed the comments raised, and we sincerely appreciate your valuable suggestions.

Regarding the code, we have made several improvements to enhance usability: (1) the README file has been updated to include instructions for automatic environment setup; (2) additional annotations have been added to the code to improve readability; (3) a code structure flowchart has been provided to help readers better understand the connections among scripts; and (4) our method has been wrapped as a Class, with a demonstration script (RunState.py) included on how to import and run it, making the code more user-friendly.

Comments from Reviewer 4

I co-reviewed this manuscript with one of the reviewers who provided the listed reports. This is part of the Nature Communications initiative to facilitate training in peer review and to provide appropriate recognition for Early Career Researchers who co-review manuscripts. Paper results are reproducible based on the published code.

Response: We sincerely thank you for co-reviewing the manuscript. Your time and effort in evaluating our work are greatly appreciated.